# HESSO: Towards Automatic Efficient and User Friendly Any Neural Network Training and Pruning

## Abstract

Structured pruning is one of the most popular approaches to effectively compress the heavy deep neural networks (DNNs) into compact sub-networks while retaining the original network performance. The existing methods suffer from multi-stage procedures along with significant engineering efforts and human expertise. The Only-Train-Once series (OTOv1-v3) has been recently proposed to resolve the many pain points by streamlining the workflow. However, the built-in sparse optimizers in the OTO series, *i.e.*, the Half-Space Projected Gradient (HSPG) family, have limitations that require hyper-parameter tuning and the implicit controls of the sparsity exploration, consequently requires intervening by human expertise. To further address such limitations, we propose a novel Hybrid Efficient Structured Sparse Optimizer (HESSO). HESSO could automatically and efficiently train a DNN within a single run to produce a high-performing sub-network. Meanwhile, it is almost tuning-free and enjoys user-friendly integration for generic training applications. To address another common issue of irreversible pruning performance collapse observed in some DNNs, we further propose a novel Corrective Redundant Identification Cycle (CRIC) to plug into HESSO for reliably identifying indispensable structures. We numerically demonstrate the efficacy of HESSO and its enhanced version HESSO-CRIC on a variety of applications ranging from computer vision to natural language processing, including large language model. The numerical results showcase that HESSO can achieve competitive performance to varying state-of-the-art benchmarks and support most DNN architectures. Meanwhile, CRIC can effectively prevent the irreversible performance collapse and further enhance the performance of HESSO on certain applications.

## 1 Introduction

Large deep neural networks (DNNs) have successfully powered a variety of applications (Ji and Chen, 2019; Zhou et al., 2024; Zhu et al., 2023). However, their typical significant time and space complexities make inference expensive and restrict deployment in resource-constrained environments. Consequently, how to compress the full DNN to the greatest extend while preserving the performance becomes essential in the many industrial and academic AI deployment pipelines. There are various model compression techniques including but not limited to pruning (Chen et al., 2021b; 2023c; Fang et al., 2023), knowledge distillation (Ko et al., 2024) and quantization (Han et al., 2015), which have been well developed in the past decades.

Structured pruning typically serves as the foremost technique to produce an optimal sub-network from a pre-defined full DNN by identifying and removing redundant structures (Gale et al., 2019; Han et al., 2015; Chen et al., 2021b; 2023c; Fang et al., 2023; Wang et al., 2024; Wu et al., 2024). Classical pruning methods focus on conducting a multi-stage procedure, requiring significant engineering efforts and expertise to manually build pruning search space, identify redundant structures, construct sub-network, and fine-tune to recover lost knowledge. To alleviate the human engineering burden, recent works (Chen et al., 2023c;b; Fang et al., 2023) have proposed pruning dependency graph to automate the pruning search space and sub-network construction. OTOv1-v2 (Chen et al., 2021b; 2023c) further unify these multi-stage components together, requiring only a single training run to directly get a compact sub-network without the need of further fine-tuning. Specifically,

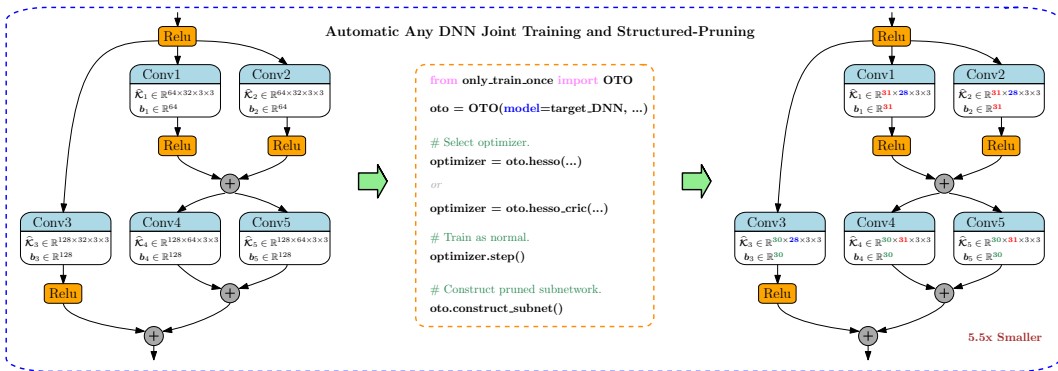

Figure 1: Automatic any DNN joint training and structured pruning experience achieved by the pruning mode of OTO along with the proposed HESSO and its enhanced HESSO-CRIC optimizer. The procedure could be applied onto varying DNN and applications, and seamlessly integrated into any training pipeline to directly produce a compact pruned sub-network without further fine-tuning.

they rely on (Dual) Half-Space Stochastic Gradient Descent (D)HSPG methods to train and prune simultaneously and have introduced a rigorous theoretical version AdaHSPG+ (Dai et al., 2023).

Although OTOv1 and OTOv2 have significantly advanced the ease of use in DNN joint training and structured pruning, they still face challenges related to the complexity of the built-in (D)HSPG methods (Chen et al., 2021b; 2023c; 2020c;a). Specifically, these methods often require substantial hyper-parameter tuning for different downstream applications and DNN architectures (Dai et al., 2023; Wu et al., 2024). Furthermore, the sparsity explorations are implicit, which requires optimization expertise, thereby diminishes the practical convenience and usability.

| | (D)HSPG | HESSO | HESSO-CRIC |
|---|---|---|---|
| **Efficiency** | ★★ | ★★★ | ★★☆ |
| **Tuning-Free** | ★ | ★★★ | ★★★ |
| **User-Friendliness** | ★ | ★★★ | ★★★ |
| **Performance** | ★★★[†] | ★★☆ | ★★★ |

[†] Under sufficient hyper-parameter tuning efforts.

Meanwhile, many modern pruning and neural architecture search methods rely on saliency scores (such as Taylor based) to identify redundant structures. However, they often suffer from performance degradation due to mistakenly identifying indispensable structures as redundant. This degradation can sometimes be irreversible due to architectural design constraints, transparency of training datasets, and the significant training resources required, posing practical challenges for their use.

To overcome these pain-points, we naturally ask, *i.e.*, how to get a joint training and pruning optimizer which is *ease-to-use*, *reliable*, *high-performing*, and *applicable* onto any DNNs and tasks.

In this work, we address this question by proposing **HESSO**: **H**ybrid **E**fficient **S**tructured **S**parse **O**ptimizer for automatic one-shot any DNN training and structured pruning. Compared to the HSPG family, HESSO offers several advantages. First, HESSO significantly simplifies the hyper-parameter setup, providing considerable practical convenience. Second, HESSO employs a progressive pruning strategy to explicitly control the sparsity exploration, making it user-friendly. Third, HESSO optionally incorporates a novel **C**orrective **R**edundancy **I**dentification **C**ycle (**CRIC**) mechanism, so-called HESSO-CRIC, which more accurately identifies redundant groups, thereby minimizing the risk of irreversible performance collapse caused by pruning indispensable structures. We now summarize our main contributions as follows.

- **Efficient Hybrid Training and Pruning Optimizer.** We propose an efficient and easy-to-use optimizer, HESSO, to enable automatic joint structured pruning and training for various model architectures and applications. HESSO progressively identifies redundant groups through flexible saliency score estimations and utilizes a hybrid training schema to effectively transfer knowledge from redundant groups to important ones, thereby maintaining the performance of the pruned model. Compared to the D(HSPG) in OTO, HESSO explicitly controls sparsity exploration and knowledge transfer, minimizes the need for hyper-parameter tuning. As a result, HESSO becomes the first optimizer to realize convenient joint DNN training and pruning to our knowledge.

- **Corrective Redundancy Identification Cycle.** We propose a novel Corrective Redundancy Identification Cycle (CRIC) to significantly improve the accuracy of redundancy identification. CRIC

addresses the approximation errors often associated with popular Taylor-based saliency scores, thereby reducing the risk of mistakenly pruning indispensable groups. CRIC employs a voting mechanism and measures the saliency scores of each group candidate using a multi-sampling approach towards the origin. CRIC is pluggable into HESSO or future joint optimizers to help to ensure reliable model performance by providing a more accurate assessment of group significance.

- **Numerical Experiments.** We validate the efficacy of HESSO and its enhanced version HESSO-CRIC across a variety of applications and model architectures. Specifically, we evaluate its performance on high-level computer vision tasks such as image classification and object detection, low-level vision tasks like super-resolution, as well as natural language processing tasks including large language models. The numerical results demonstrate that HESSO performs competitively, and in many cases, exceeds the state-of-the-art benchmarks, offering significant practical convenience. Additionally, CRIC effectively mitigates the issues of irreversible collapse in pruned models, especially in challenging cases, further showcasing its utility.

## 2  RELATED WORKS OF AUTOMATED STRUCTURED PRUNING

In this section, we provide a brief literature review on automatic structured pruning, while additional reviews on knowledge transfer and DNN architecture optimization can be found in Appendix A.

**General Pruning Procedures.**    Structured pruning aims to compress DNNs by removing unnecessary structures while maintaining performance (Han et al., 2015; Wen et al., 2016). The general procedure typically involves: (*i*) training a full model; (*ii*) identifying and removing redundant structures to construct a slimmer DNN based on various criteria (Lin et al., 2019; He et al., 2018a; Wen et al., 2016; Li et al., 2020b; Zhuang et al., 2020; Chen et al., 2017; 2018; 2021a; 2020b; Gao et al., 2020; Zhuang et al., 2020; Meng et al., 2020; Yang et al., 2019; Zhou et al., 2019; van Baalen et al., 2020; Frankle and Carbin, 2018); and (*iii*) retraining the pruned model to recover any accuracy lost during pruning. These methods often require a complex and time-consuming process, involving multiple training iterations and significant domain knowledge to manually handle each step.

**Automated Pruning Given Pre-defined Search Space.**    To resolve the pain-points of human interventions, automated pruning is raising interests from different perspectives. Given a predefined search space, AMC (He et al., 2018b) employs reinforcement learning agents to automatically determine the optimal pruning ratio. EagleEye (Li et al., 2020a) further introduces a sub-network evaluation scheme based on adaptive batch normalization, which can be integrated into AMC. OFA (Cai et al., 2019) automates the generation of sub-networks for different hardware platforms in a single process. While these approaches yield impressive performance, their application is limited to predefined search spaces. Moreover, AMC incurs additional training costs for its reinforcement learning agent. OFA's training procedure is complex and heavy to adopt all sub-networks. It also requires knowing the optimal training procedure for the largest super-network in advance to ensure the performance, which makes practical adoption less convenient.

**Automated Pruning Over Any DNNs.**    On the other hand, automatically pruning arbitrary models without prior knowledge of the search space remained a significant challenge. Recent methods, such as OTO (Chen et al., 2021b; 2023c;b) and DepGraph (Fang et al., 2023), have made progress in automating the structured pruning process for general DNNs via dependency graph analysis. Subsequent works like (Wang et al., 2024) and (Ren et al., 2024) automates pruning over ONNX models. ATO (Wu et al., 2024) introduces ControlNet upon OTOv2. Among these, OTO offers a one-shot joint training and pruning framework that can seamlessly integrate into various training processes to produce high-performing sub-networks in a single run. While these automated approaches have significantly improved user convenience, end-users still face significant challenges with hyper-parameter tuning and the sparse optimization expertise required to calibrate OTO's built-in HSPG family (Chen et al., 2020c; Dai et al., 2023). Furthermore, some DNNs contain indispensable structures, the pruning of which leads to irreversible performance degradation. Identifying these critical structures remains an open problem that is often handled manually on a case-by-case basis, complicating practical use. In this work, we tackle these pain points to propose an efficient, tuning-free, and user-friendly joint training and pruning optimizer HESSO along with its enhanced version HESSO-CRIC to reliably identify indispensable structures to ensure the performance.

---

**Algorithm 1** HESSO: Hybrid Efficient Structured Sparsity Optimizer

---

1: **Input.** Initial variable $\boldsymbol{x}_0$, learning rate $\alpha$, warm-up steps $T_w$, pruning periods $P$, period length $T_p$, target group sparsity level $K$, and variable partition $\mathcal{G} = \mathcal{G}_{\text{prunable}} \bigcup \mathcal{G}_{\text{unprunable}}$.

2: Warm up $T_w$ steps via SGD or its variants, *e.g.*, AdamW.

3: Initialize redundant groups $\mathcal{G}_R \leftarrow \varnothing$ and important groups $\mathcal{G}_I \leftarrow \mathcal{G}$.

4: Compute sparsity level for each pruning period $\widehat{K} := K/T_p$.

5: **for** each pruning period $p = 0, 1, \cdots, P - 1$ **do**

6:      Pickup $\widehat{\mathcal{G}}_p$ in $\mathcal{G}_I$ with $\widehat{K}$-least saliency scores.

7:      Update $\mathcal{G}_R \leftarrow \mathcal{G}_R \cup \widehat{\mathcal{G}}_p$ and $\mathcal{G}_I \leftarrow \mathcal{G}_I/\widehat{\mathcal{G}}_p$.

8:      **for** $t = 0, 1, \cdots, T_p - 1$ **do**

9:          Compute trial iterate $\widehat{\boldsymbol{x}}_{t+1} \leftarrow \boldsymbol{x}_t - \alpha_t \nabla f(\boldsymbol{x}_t)$.

10:          Compute transferring penalty ratio $[\boldsymbol{\gamma}_t]_g \leftarrow \frac{T_p - t - 1}{T_p - t} \frac{\|[\boldsymbol{x}_t]_g\|}{\|[\widehat{\boldsymbol{x}}_{t+1}]_g\|}$ for each $g \in \widehat{\mathcal{G}}_p$.

11:          Update redundant group variables $[\boldsymbol{x}_{t+1}]_{\widehat{\mathcal{G}}_p} \leftarrow [\boldsymbol{\gamma}_t]_{\widehat{\mathcal{G}}_p} [\widehat{\boldsymbol{x}}_{t+1}]_{\widehat{\mathcal{G}}_p}$.

12:          Update important group variables $[\boldsymbol{x}_{t+1}]_{\mathcal{G}_I} \leftarrow [\widehat{\boldsymbol{x}}_{t+1}]_{\mathcal{G}_I}$.

13:      **end for**

14: **end for**

15: Training important group variables till convergence.

16: **Return** the final iterate $\boldsymbol{x}^*_{\text{HESSO}}$.

---

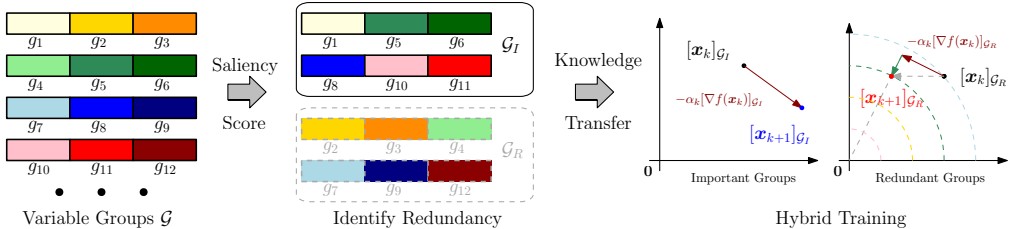

Figure 2: HESSO uses saliency scores to periodically identify redundant groups $\mathcal{G}_R$ from the group set $\mathcal{G}$ and marks the remaining groups as important groups $\mathcal{G}_I$. A knowledge transfer mechanism is proceeded by employing hybrid training strategies onto $\mathcal{G}_R$ and $\mathcal{G}_I$. In particular, the variables in $\mathcal{G}_R$ are progressively projected onto zeros after gradient descent. The important variables are kept training via gradient descent to migrate the impact of redundant project onto the objective function.

## 3 HESSO

Given a target DNN which variables and architecture to be optimized, HESSO formulates a constrained structured sparsity optimization problem upon the parameter groups $\mathcal{G}$ as (1).

$$\underset{\boldsymbol{x} \in \mathbb{R}^n}{\text{minimize}} \ f(\boldsymbol{x}), \quad \text{s.t. Cardinality}\{g \in \mathcal{G} | [\boldsymbol{x}]_g = 0\} = K, \tag{1}$$

where we seek to yield group sparsity over the prunable variables with the target sparsity level as $K$. The parameter groups can be pruning zero-invariant groups determined through the pruning dependency graph analysis or other general group formats (Chen et al., 2023c;b).

During the optimization process, HESSO begins with a warm-up phase, where the variables are trained using gradient descent or its variants. The purpose of the warm-up stage is to collect gradient information and guide the DNN into a relatively favorable region for convergence. Following this, HESSO performs progressive pruning by periodically identifying redundant parameter groups based on predefined saliency scores. Throughout the progressive pruning phase, HESSO gradually forgets the knowledge in the redundant groups while the remaining important groups continue training, thereby facilitating the transfer and recapture of knowledge. We refer to this approach as hybrid training, where distinct training strategies are applied to different groups. Finally, once all redundant groups are identified and projected onto zero, the remaining important groups continue to be trained until final convergence. The main procedure is outlined in Algorithm 1.

### 3.1 SALIENCY SCORE

After warming up $T_w$ steps in Algorithm 1, HESSO has typically collected reasonable information regarding the gradient and the iterate. It then starts to identify redundant groups upon the target group sparsity level $K$ to partition the groups $\mathcal{G}$ into important group set $\mathcal{G}_I$ and redundant group set $\mathcal{G}_R$, *i.e.*, $\mathcal{G}_I \bigcup \mathcal{G}_R = \mathcal{G}$ and $|\mathcal{G}_R| = K$. HESSO achieves it by periodically measuring the importance of each parameter group $g \in \mathcal{G}$. To begin, we initialize the important group set as the whole group set $\mathcal{G}_I \leftarrow \mathcal{G}$, and the redundant group set as empty $\mathcal{G}_R \leftarrow \varnothing$. Given a pre-defined pruning periods $P$, we identify $\widehat{K} \leftarrow K/P$ important groups to designate as redundant during each period. The redundant groups are the ones with bottom-$\widehat{K}$ saliency scores.

$$
\begin{aligned}
\mathcal{G}_R &\leftarrow \mathcal{G}_R \bigcup \underset{g \in \mathcal{G}_I}{\text{Bottom-}\widehat{K}} \ \text{SaliencyScore}([\boldsymbol{x}]_g, [\nabla f(\boldsymbol{x})]_g) \\
\mathcal{G}_I &\leftarrow \mathcal{G}_I / \underset{g \in \mathcal{G}_I}{\text{Bottom-}\widehat{K}} \ \text{SaliencyScore}([\boldsymbol{x}]_g, [\nabla f(\boldsymbol{x})]_g)
\end{aligned}
\tag{2}
$$

The selection of the saliency score in HESSO is flexible and can be tailored to different purposes. By default, we consider the categories presented in Appendix D.

### 3.2 HYBRID TRAINING IN HESSO

After identifying the redundant groups in Section 3.1, the next step is to project these groups onto zero and transfer their knowledge to the important groups, ensuring the pruned model maintains its performance. This is achieved through a hybrid training schema.

For the redundant groups $\mathcal{G}_R$, we progressively and uniformly push their parameters towards zero. This process is detailed in line 7-8 in Algorithm 1 and decipted in Figure 2. The goal is to ensure that the parameters in the redundant groups become zero after $T_p$ steps. During this penalization process, there is a risk of forgetting the knowledge contained in the redundant groups, which may manifest as a degradation in the objective function's value. To mitigate this, we employ a standard optimization method, such as vanilla SGD or its variants like Adam, on the important groups $\mathcal{G}_I$. This step aims to continue optimizing the objective function $f$ and preserve the model's performance despite the pruning of redundant groups. By maintaining the optimization of the important groups, the knowledge lost from the redundant groups can be transferred and compensated for, ensuring that the pruned model remains effective.

Next, we provide brief intuitive comparisons of HESSO against two popular pruning algorithms.

**Minimize tuning efforts compared to DHSPG.** DHSPG in OTOv2 involves significant hyper-parameter tuning to adjust parameters for sparsity exploration. This tuning often requires domain-specific knowledge, as the appropriate settings can vary depending on the particular application or dataset. This requirement can make DHSPG more complex and less accessible, particularly for practitioners without extensive expertise in hyper-parameter and sparse optimization. Contrarily, HESSO offers more explicit control over sparsity exploration. The pruning process in HESSO is regulated by the pruning periods $P$ and the period length $T_P$, which determine the pace and extent of the pruning procedure. This structured approach simplifies the process, making it easier to manage.

**Architecture-agnostic computational invariance compared to ResRep and SliceGPT.** ResRep (Ding et al., 2021b) and SliceGPT (Ashkboos et al., 2024) are proposed to preserve computational invariance, *i.e.*, making pruned and full models produce similar outputs, for CNNs and transformers, respectively. However, they are architecture specific, requires additional efforts, such as injecting additional layers in SliceGPT and computing reset gradients in ResRep. The knowledge transfer in HESSO similarly seeks to maintain computational invariance but does so by preserving objective function levels. In addition, HESSO is architecture-agnostic, efficient and user-friendly, demonstrating both scalability and versatility compared with ResRep and SliceGPT.

As a result, HESSO is generally easier to use and more adaptable to various applications, as it significantly reduces the need for extensive tuning and specialized knowledge. The design of hybrid training for knowledge transfer effectively promotes the performance of pruned model. They make HESSO a more user-friendly and efficient option for achieving structured sparsity in models, allowing for more straightforward and consistent application across different tasks and domains.

## 3.3 Approximation Errors of Salience Scores

Although HESSO could tackle most DNNs and applications, it may sometimes yield unsatisfactory results when the target DNN possesses certain indispensable structures, defined as follows.

**Definition 3.1** (Indispensable structure). Given a deep neural network $\mathcal{M}$, a minimally removal structure is called indispensable if removing it from $\mathcal{M}$ would cause significant performance degradation, which can not be recovered given user resources. In particular, we say a minimally removal structure as $\epsilon$-indispensable associated with an objective $f$ if pruning the variables $[\boldsymbol{x}]_g \to \boldsymbol{0}$ deteriorates $f$ at least $\epsilon$, i.e., $f(\boldsymbol{x}|[\boldsymbol{x}]_g \to \boldsymbol{0}) \geq f(\boldsymbol{x}) + \epsilon$ for a minimization optimization problem. The degradation $\epsilon$ can not be recovered by (i) keeping training $\mathcal{M}$, (ii) the training cost such as GPU days exceeding user budget, or (iii) the training receipt for $\mathcal{M}$ is black-box and hard to be reproduced.

The origin of indispensable structures varies. One reason may be due to architectural design issues where certain layers in $\mathcal{M}$ play more critical roles than others and are very sensitive to any modifications, as exemplified by a low-level vision benchmark in Section 4.2. Another reason could be the learning strategy. For instance, in large language models (LLMs), it has been observed that knowledge is unevenly distributed across different layers (Chen et al., 2023a). Removing any of these structures could result in an irreversible collapse of the DNN's performance.

**Salience score approximation errors.** The existing saliency scores might fail to identify these indispensable components accurately. As described in Appendix D, they are typically designed to approximate the impact of projecting groups of variables to zero over the objective function. Such approximations, for example, perhaps the most commonly used Taylor importance scores, are more accurate when the iterate is close enough to the origin point.

**Theorem 3.2** (Approximation error of Taylor importance). *Suppose the gradient and second-order derivative of $f$ are bounded. Use first-order $m^L$ and second-order $m^Q$ Taylor approximations to measure the function value $f$ after pruning $g \in \mathcal{G}$, i.e., $[\boldsymbol{x}]_g \to \boldsymbol{0}$. Let $\boldsymbol{s}$ satisfy $[\boldsymbol{s}]_{\mathcal{G}/g} = [\boldsymbol{0}]_{\mathcal{G}/g}$ and $[\boldsymbol{s}]_g = -[\boldsymbol{x}]_g$, Then the approximation error bound $|f(\boldsymbol{x}+\boldsymbol{s})-m^L(\boldsymbol{x}+\boldsymbol{s})|$ and $|f(\boldsymbol{x}+\boldsymbol{s})-m^Q(\boldsymbol{x}+\boldsymbol{s})|$ are proportional to $\mathcal{O}(\|[\boldsymbol{x}]_g\|^2)$ and $\mathcal{O}(\|[\boldsymbol{x}]_g\|^3)$, respectively.*

However, during realistic training and pruning, this requirement is usually not met. As stated in Theorem 3.2, the approximation error bounds increase proportionally with $\|[\boldsymbol{x}]_g\|$, indicating that the further the distance from the origin, the larger the approximation error. As a result, this can lead to the false positively pruning of indispensable structures, which in turn causes performance issues.

## 3.4 Corrective Redundancy Identification Circle

To address the limitations discussed in Section 3.3, we propose a novel Corrective Redundant Identification Cycle (CRIC). This method aims to more reliably identify redundant structures within the target DNN, even when indispensable structures are present. The CRIC mechanism can be seamlessly integrated into HESSO, enhancing its ability to accurately discern which parts of the model can be pruned without compromising performance.

To mitigate the issue of false positive redundant predictions caused by the approximation error, such as Taylor expansion, CRIC measures the saliency score of redundant group candidates multiple times along the projection to the origin. Unlike the greedy approach in HESSO, CRIC incorporates a corrective cycle mechanism. This mechanism iteratively promotes groups as redundant and tracks the outlier groups. The cycle terminates when the redundancy prediction is deemed reliable, i.e., no outlier appearance is detected. The final output is a set of redundant groups $\mathcal{G}_R$ with the bottom-K overall saliency scores. This approach significantly reduces false positive redundant identifications and addresses the failure cases of HESSO, as demonstrated numerically in Section 4.

In Algorithm 2, we utilize a violating group set $\mathcal{V}$ to track outlier or violating groups, which are more redundant or deviate from the current redundant group prediction. $\mathcal{V}$ is initialized with the group set having the bottom-K saliency scores (see line 3). A historical set $\mathcal{H}$ is also used to track groups whose saliency scores have been fully exploited through multiple sampling along the projection to the origin. This set is initialized as empty $\varnothing$, as shown in line 4.

When the violating set is fairly large, i.e., $|\mathcal{V}| > \mathcal{T}$ with $\mathcal{T}$ as a predefined terminating tolerance which is by default as empty set, i.e., $\mathcal{T} = \varnothing$, we progressively project these violating groups onto

---

**Algorithm 2** Corrective Redundant Identification Cycle (CRIC)

---

1: **Input.** Trainable variable $\boldsymbol{x}$, learning rate $\alpha$, termination tolerance $\mathcal{T}$, target group sparsity $K$, sample steps $T$, and prunable variable partition $\mathcal{G}$.
2: Initialize $\mathcal{S}$ to store saliency scores for each $g \in \mathcal{G}$.
3: Initialize violating group set $\mathcal{V}$

$$\mathcal{V} \leftarrow \{g : g \in \mathcal{G} \text{ with bottom-K saliency scores}\}. \tag{3}$$

4: Initialize historical set $\mathcal{H} \leftarrow \mathcal{V}$.
5: **while** $|\mathcal{V}| \leq \mathcal{T}$ **do**
6:     Initialize trial violating group set $\widehat{\mathcal{V}} \leftarrow \varnothing$.
7:     Initialize $\alpha_0 \leftarrow \alpha$, $\lambda_0 \leftarrow \lambda$, and $\boldsymbol{x}_0 \leftarrow \boldsymbol{x}$.
8:     **for** $t = 0, 1, \cdots, T-1$ **do**
9:         Compute gradient of $f$ over $\boldsymbol{x}_t$ as $\nabla f(\boldsymbol{x}_t)$.
10:         Compute trial $\tilde{\boldsymbol{x}}_{t+1} \leftarrow \boldsymbol{x}_t - \alpha_t \nabla f(\boldsymbol{x}_t)$.
11:         Penalize variables in the violating set $[\boldsymbol{x}_{t+1}]_{\mathcal{V}} \leftarrow \frac{T-t-1}{T-t} \frac{[\boldsymbol{x}_t]_{\mathcal{V}}}{\|[\tilde{\boldsymbol{x}}_{t+1}]_{\mathcal{V}}\|}$.
12:         Compute saliency scores of $\mathcal{G}$ and collect into $\mathcal{S}$.
13:         Update trial set $\widehat{\mathcal{V}}$ if new violating groups appear.

$$\widehat{\mathcal{V}} \leftarrow \widehat{\mathcal{V}} \cup \{g : g \in \mathcal{G} \text{ with bottom-K scores}\}/\mathcal{V}. \tag{4}$$

14:         Update penalty $\lambda_t$ and learning rate $\alpha_t$.
15:     **end for**
16:     Update violating set $\mathcal{V} \leftarrow \widehat{\mathcal{V}}/\mathcal{H}$.
17:     Update historical set $\mathcal{H} \leftarrow \mathcal{H} \cup \mathcal{V}$.
18: **end while**
19: Set redundant set $\mathcal{G}_R$ upon saliency score collection $\mathcal{S}$.

$$\mathcal{G}_R \leftarrow \{g : g \text{ with bottom-K scores in } \mathcal{S}\} \tag{5}$$

20: **Return.** Identified redundant group set $\mathcal{G}_R$ and important group set $\mathcal{G}_I$ as $\mathcal{G}/\mathcal{G}_R$.

---

zero. By default, saliency score sampling points are uniformly distributed along the projection process. Groups with lower importance scores that have not been visited in $\mathcal{H}$ are added to a newly constructed violating set $\hat{\mathcal{V}}$ for the next corrective cycle. The corrective cycling algorithm continues until violating instances rarely appear, *i.e.*, $|\mathcal{V}| \leq \mathcal{T}$, see line 5.

Theorem 3.3 guarantees that CRIC terminates within a finite number of iterations, preventing endless loops and executing efficiently. We provided detailed proof for Theorem 3.3 in Appendix C. Furthermore, Corollary 3.4 provides an upper bound on the number of cycles required by CRIC, ensuring a practical and efficient pruning process.

**Theorem 3.3** (Finite termination of CRIC). *The corrective redundancy identification cycle (Algorithm 2) terminates within a finite number of steps for any terminating tolerance $\mathcal{T}$.*

**Corollary 3.4** (Upper bounds of cycle numbers). *Given the terminating tolerance $\mathcal{T}$, the CRIC terminates with no more $(|\mathcal{G}| - K)/\max\{\mathcal{T}, 1\}$ cycles.*

Once the corrective cycles terminate, the saliency scores obtained are deemed reliable. At this point, the redundant set $\mathcal{G}_R$ is constructed based on these reliable saliency scores, as indicated in line 19. This set of redundant groups is then returned for further use, such as hybrid training in HESSO (as detailed in Algorithm 1). For simplicity, the HESSO variant that utilizes CRIC for identifying redundant groups is referred to as HESSO-CRIC throughout the paper (as outlined in Algorithm 3). This naming convention helps distinguish the variant from the original HESSO method, emphasizing the integration of the corrective cycle mechanism to enhance the reliability of the pruning process.

## 4 NUMERICAL EXPERIMENTS

We numerically demonstrate the efficacy of HESSO across a wide range of applications, from low-level vision tasks such as super-resolution (Zhou et al., 2024), to high-level vision tasks like image

---

**Algorithm 3** HESSO-CRIC

---

1: **Input.** trainable variable $x_0$, learning rate $\alpha$, warm-up steps, $T_w$, and hybrid training steps $T_h$.
2: Warm-up for $T_w$ steps via SGD or its variants.
3: Use CRIC in Algorithm 2 to get redundant and important group sets $\mathcal{G}_R$ and $\mathcal{G}_I$.
4: *Hybrid Training for Knowledge Transfer.*
5: **for** $t = 0, 1, \cdots, T_h$ **do**
6:     Compute trial iterate $\widehat{x}_{t+1} \leftarrow x_t - \alpha_t \nabla f(x_t)$.
7:     Compute transferring penalty ratio $[\gamma_t]_g \leftarrow \frac{T-t-1}{T-t} \frac{\|[x_t]_g\|}{\|[\widehat{x}_{t+1}]_g\|}$ for each $g \in \mathcal{G}_R$.
8:     Update redundant group variables $[x_{t+1}]_{\mathcal{G}_R} \leftarrow [\gamma_t]_{\mathcal{G}_R} [\widehat{x}_{t+1}]_{\mathcal{G}_R}$.
9:     Update important group variables $[x_{t+1}]_{\mathcal{G}_I} \leftarrow [\widehat{x}_{t+1}]_{\mathcal{G}_I}$.
10: **end for**
11: Keep training variables in important groups till convergence.
12: **Output.** The final iterate $x^*$.

---

classification (He et al., 2016) and object detection (Shi et al., 2020), as well as natural language processing tasks such as question answering (Rajpurkar et al., 2016) and the popular foundational large language models (Ding et al., 2023). The architectures used in these experiments encompass a variety of CNN benchmarks (Chen et al., 2023c) and transformers (Vaswani et al., 2017). These experiments involve training either from scratch or using a pre-trained checkpoint (when available) to validate the versatility of HESSO-(CRIC).

## 4.1 RECOMMENDED EXPERIMENTAL SETUP

Table 1: Recommended hyper-parameters and training strategies for HESSO and HESSO-CRIC.

| Hyper-parameter | Type | Recommended Setup |
|---|---|---|
| Optimizer variant | HESSO-(CRIC) | Inherit as the baseline optimizer. Currently support {SGD, Adam, AdamW}. |
| Group sparsity | HESSO-(CRIC) | Set upon the target pruned model size. If all variables could be pruned, the pruned model size could be approximately equal as quadratic of the density level. In addition, a randomly pruned model could be obtained by OTO's APIs. |
| First-order momentum | HESSO-(CRIC) | Inherit as the baseline optimizer's first-order momentum. |
| Second-order momentum | HESSO-(CRIC) | Inherit as the baseline optimizer's second-order momentum. |
| Weight-decay | HESSO-(CRIC) | Inherit as the baseline optimizer's weight-decay. |
| Initial learning rate | HESSO-(CRIC) | Inherit as the baseline optimizer's initial learning rate. |
| Salience Score Criteria | HESSO-(CRIC) | By default equally considering the scores in Section 3.1. |
| Start pruning step | HESSO-(CRIC) | Set up as 1/10 of total training steps. |
| Pruning steps | HESSO-(CRIC) | Set up as 1/10 of total training steps. |
| Pruning periods | HESSO | Empirically suggest to set as 10. |
| Sampling steps | HESSO-CRIC | Empirically suggest to set as 10. |
| Learning rate scheduler | Training | Inherit as the baseline training, yet might need adjustments in some application to ensure the model after reaching target group sparsity is sufficiently trained under relatively large learning rate. |
| Total training steps | Training | Inherit as the baseline training and adjust upon the learning rate scheduler. |
| Start training from scratch or pre-training checkpoint | Training | Both are supported. For better performance, recommend to start from pretraining checkpoint if available. |

We recommend the following hyperparameter configurations for HESSO and HESSO-CRIC across varying applications and DNN architectures. For the target DNN to be trained and compressed, end-users likely already have a well-established training pipeline that enables the DNN to achieve high performance. To ensure ease of use, we suggest inheriting the hyperparameters in HESSO and HESSO-CRIC from the baseline training schema wherever there is overlap, such as in optimization variants and first- and second-order momentums.

This inheritance strategy should also be applied to other hyperparameters related to the training pipeline, such as training steps and learning rate schedules, though some slight adjustments may be needed for some applications. Specifically, adjustments may be needed due to the hybrid training process. We recommend beginning pruning at 1/10 of the total training steps and completing progressive pruning over another 1/10 of the total training steps. Because of the hybrid training stage, the learning rate schedule might require modification to ensure the DNN is sufficiently trained at a reasonably high learning rate after reaching the target group sparsity level.

Additionally, HESSO and HESSO-CRIC support training either from scratch or from a pre-trained checkpoint. For better performance and faster convergence, we recommend starting from a pre-

trained status if such a checkpoint is available. We summarize the recommended hyperparameter selections and training strategies in Table 1 Appendix **??**. Remark that better hyperparameter setups or training strategies may exist for specific domain tasks to achieve superior performance. For the remainder of the manuscript, we conduct experiments according to the above recommended criteria, unless otherwise specified. All experiments were conducted on an A100 GPU with 80GB memory .

## 4.2 SUPER RESOLUTION

We first selected the popular CARN architecture (Ahn et al., 2018) for the super-resolution task with a scaling factor of two, referred to as CARNx2. The benchmark DIV2K dataset (Agustsson and Timofte, 2017) was used for training, while Set14 (Zeyde et al., 2010), B100 (Martin et al., 2001), and Urban100 (Huang et al., 2015) datasets were employed for evaluation. Initially, we utilized OTO's pruning dependency analysis to identify minimally removable structures and partitioned the trainable variables into pruning-zero-invariant groups. However, directly applying DHSPG or HESSO led to significant performance degradation that was not reversible. This issue arose due to the architectural design, where the penultimate convolutional layer plays a crucial role in producing satisfactory visual results, making it a indispensable structure. Pruning this layer caused the remaining filters to fail in generating reasonable visual outcomes. However, the saliency score deems them as redundant due to significant approximation errors, a greedy identification schema fails to avoid pruning such essential structures, resulting in irreversible performance collapse.

Table 2: Structurally pruning CARNx2.

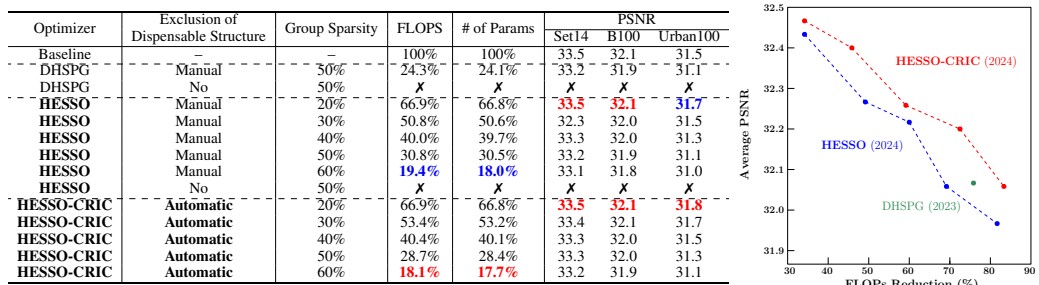

| Optimizer | Exclusion of Dispensable Structure | Group Sparsity | FLOPS | # of Params | PSNR Set14 | B100 | Urban100 |
|---|---|---|---|---|---|---|---|
| Baseline | – | – | 100% | 100% | 33.5 | 32.1 | 31.5 |
| DHSPG | Manual | 50% | 24.3% | 24.1% | 33.2 | 31.9 | 31.1 |
| DHSPG | No | 50% | ✗ | ✗ | ✗ | ✗ | ✗ |
| HESSO | Manual | 20% | 66.9% | 66.8% | **33.5** | **32.1** | **31.7** |
| HESSO | Manual | 30% | 50.8% | 50.6% | 32.3 | 32.0 | 31.5 |
| HESSO | Manual | 40% | 40.0% | 39.7% | 33.3 | 32.0 | 31.3 |
| HESSO | Manual | 50% | 30.8% | 30.5% | 33.2 | 31.9 | 31.1 |
| HESSO | Manual | 60% | **19.4%** | **18.0%** | 33.1 | 31.8 | 31.0 |
| HESSO | No | 50% | ✗ | ✗ | ✗ | ✗ | ✗ |
| HESSO-CRIC | Automatic | 20% | 66.9% | 66.8% | **33.5** | **32.1** | **31.8** |
| HESSO-CRIC | Automatic | 30% | 53.4% | 53.2% | 33.4 | 32.1 | 31.7 |
| HESSO-CRIC | Automatic | 40% | 40.4% | 40.1% | 33.3 | 32.0 | 31.5 |
| HESSO-CRIC | Automatic | 50% | 28.7% | 28.4% | 33.3 | 32.0 | 31.3 |
| HESSO-CRIC | Automatic | 60% | **18.1%** | **17.7%** | 33.2 | 31.9 | 31.1 |

OTOv2 (Chen et al., 2023c) manually excluded these indispensable structures from pruning. However, this manual identification is time-consuming and requires expert knowledge. To address this, we directly applied HESSO-CRIC to CARN and observed that it automatically identified these crucial structures as important groups, leading to a successfully high-performing pruned model. As shown in Table 2, when manually excluding indispensable structures, both DHSPG and HESSO significantly reduced FLOPs and parameters by approximately 33% to 80%, with negligible PSNR degradation. HESSO-CRIC achieved a better trade-off between FLOP reduction and PSNR, as demonstrated by exhibiting the frontier curve under varying pruning ratios. Visual examples shown in Figure 7 further cross-verify the effective performance preservation by our approaches.

## 4.3 IMAGE CLASSIFICATION

We then conduct on the benchmark ResNet50 (He et al., 2016) on ImageNet. As displayed in Figure 3, HESSO-CRIC roughly exhibits a Pareto frontier in terms of top-1 accuracy and FLOPs reduction under various group sparsities from 40% to 70%. HESSO and DHSPG perform competitively in this application. Meanwhile, all of them could produce structurally pruned sub-networks associated with smaller size, fewer FLOPs, and higher accuracy compared to most of the existing approaches (Huang and Wang, 2018; Zhou et al., 2019; Ding et al., 2021a; Yang et al., 2019; You et al., 2019; Zhou et al., 2019). These results well validate the efficacy of the newly proposed joint pruning and training optimizer on this popular structured pruning benchmark.

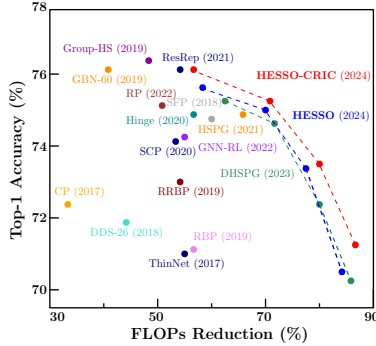

Figure 3: ResNet50 on ImageNet.

We further employ HESSO-(CRIC) to structurally prune a pretrained OFA network (Cai et al., 2019) on the benchmark ImageNet (Deng et al., 2009). The OFA network was pro- duced by searching from a Mo-

Table 3: Structurally pruning MobileNet Search Space.

| Method | # of Params (M) | MACs (M) | Top Acc-1 (%) |
|---|---|---|---|
| OFA$_{LARGE}$ # 75 (Cai et al., 2019) | 9.14 | 595 | 80.0 |
| MobileNetV2 (Sandler et al., 2018) | 3.4 | 300 | 72.0 |
| MobileNetV3-Large (Howard et al., 2019) | 5.4 | 219 | 75.2 |
| OFA # 75 (Cai et al., 2019) | 5.81 | 230 | 76.9 |
| **HESSO** | 5.60 | 220 | 78.2 |
| **HESSO-CRIC** | 5.71 | 225 | **78.6** |

bileNetV3 based super-network and could achieve 80.0% top-1 test accuracy on ImageNet. We find that both HESSO-(CRIC) could effectively discover pruned sub-networks which similar size and MACs while with higher performance than other OFA networks, *i.e.*, 78.6% and 78.2% versus 76.9% testing accuracy.

## 4.4 LARGE LANGUAGE MODEL

Finally, we evaluated HESSO-(CRIC) on large language models (LLMs). Since both HESSO and HESSO-CRIC utilize full gradient information, we focused on LLMs with fewer than 3 billion parameters, such as the representative Phi-2-2.7B (Microsoft, 2023), to ensure that a single 80GB GPU is sufficient, without requiring tensor parallelism (Ding et al., 2023). Our experimental setup followed that of LoRAShear (Chen et al., 2023a).

Table 4: HESSO-CRIC over Phi-2-2.7B.

| Pruning Ratio | Method | BoolQ | PIQA | HellaSwag | WinoGrande | ARC-e | ARC-c | OBQA | Average |
|---|---|---|---|---|---|---|---|---|---|
| Baseline | Phi-2-2.7B | 83.30 | 79.11 | 73.82 | 75.77 | 80.05 | 54.18 | 51.40 | 71.09 |
| Ratio = 20% | SliceGPT (Ashkboos et al., 2024) | 68.56 | 74.16 | 61.22 | 67.56 | 70.20 | 41.04 | 38.80 | 60.22 |
| | LLM-Pruner (Ma et al., 2023) | 61.28 | 62.79 | 36.79 | 53.12 | 52.23 | 31.06 | 30.00 | 46.75 |
| | LoraShear (Chen et al., 2023a) | 62.29 | 68.12 | 45.28 | 58.8 | 61.91 | 32.42 | 34.00 | 51.81 |
| | LoraPrune (Zhang et al., 2023) | 57.22 | 67.79 | 45.1 | 54.85 | 61.87 | 35.15 | 33.80 | 50.83 |
| | **HESSO-CRIC** | 69.67 | 74.37 | 62.27 | 66.54 | 72.30 | 41.44 | 38.20 | **60.67** |
| Ratio = 25% | SliceGPT (Ashkboos et al., 2024) | 63.70 | 71.49 | 57.72 | 66.46 | 65.86 | 38.99 | 39.80 | 57.71 |
| | LLM-Pruner (Ma et al., 2023) | 62.26 | 60.55 | 33.86 | 51.07 | 47.81 | 30.63 | 28.80 | 45.00 |
| | LoraShear (Chen et al., 2023a) | 62.17 | 64.85 | 41.27 | 55.56 | 56.52 | 30.46 | 31.80 | 48.95 |
| | LoraPrune (Zhang et al., 2023) | 62.54 | 64.69 | 40.19 | 52.33 | 56.02 | 33.62 | 32.40 | 48.83 |
| | **HESSO-CRIC** | 67.06 | 73.77 | 58.51 | 65.18 | 70.66 | 38.60 | 38.00 | **58.74** |
| Ratio = 30% | SliceGPT (Ashkboos et al., 2024) | 38.17 | 61.04 | 42.05 | 60.38 | 50.80 | 28.07 | 31.2 | 44.53 |
| | LLM-Pruner (Ma et al., 2023) | 62.11 | 59.36 | 32.27 | 51.54 | 44.07 | 30.03 | 29.8 | 44.17 |
| | LoraShear (Chen et al., 2023a) | 62.17 | 63.22 | 39.25 | 57.14 | 51.77 | 28.58 | 30.00 | 47.45 |
| | LoraPrune (Zhang et al., 2023) | 62.29 | 63.10 | 35.86 | 51.62 | 51.43 | 31.74 | 32.40 | 46.92 |
| | **HESSO-CRIC** | 67.61 | 72.14 | 53.11 | 62.75 | 62.74 | 34.81 | 36.20 | **55.62** |

We observed that without conducting a knowledge distribution analysis and manually skipping cer- tain layers from pruning, as LoRAShear (Chen et al., 2023a) did, HESSO often led to an irreversible performance collapse. This is because knowledge in LLMs is unevenly distributed across layers due to the learning strategy. The saliency scores calculated upon the pretraining weights may fail to identify essential structures, making it difficult to differentiate between indispensable components and those that could be pruned. As a result, pruning such critical structures severely degrades the model's performance, making recovery with limited resources nearly impossible.

HESSO-CRIC was able to automatically bypass these crucial structures, enabling effective and suc- cessful pruning. We then compared with SliceGPT (Ashkboos et al., 2024), LLM-Pruner (Ma et al., 2023), LoraShear (Chen et al., 2023a) and LoraPrune (Zhang et al., 2023) across several popular benchmarks. Our findings indicate that HESSO-CRIC consistently outperforms them at varying pruning ratios, with performance improvements becoming more pronounced as the pruning ratio increases. This is because LLM-Pruner, LoRA-Prune, and LoRAShear are LoRA-based techniques. Lora primarily focuses on fine-tuning well-trained models and is less effective in capturing knowl- edge for underfitted models, such as pruned LLMs.

## 5 CONCLUSION

In this work, we introduced HESSO-(CRIC), a novel Hybrid Efficient Structured Sparse Optimizer tailored for pruning deep neural networks while preserving performance. By combining a hybrid training strategy with explicit, progressive pruning control, and the Corrective Redundant Identi- fication Cycle (CRIC), HESSO-(CRIC) effectively tackles challenges such as tuning efforts, user difficulty, and irreversible performance degradation. Our experiments across diverse domains show that it not only competes with but often surpasses state-of-the-art methods.

## REPRODUCIBILITY STATEMENT

The theorems and experimental results could be fully reproduced. In particular, we provide the code base to reproduce the experimental results as supplementary materials. Meanwhile, we provide the proof for the main theorem in Appendix C.

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

## A  MORE RELATED WORKS

**Knowledge Transfer.**   To retain the performance of the pruned sub-network, HESSO-(CRIC) incorporates a knowledge transfer mechanism through a hybrid training schema. This approach differs from prior methods, which explicitly use knowledge distillation from unpruned models to preserve information in pruned models. Existing techniques typically require expensive computations that involve both pruned and unpruned models, whether by processing logits (Lagunas et al., 2021) or the hidden activations of intermediate layers (Xia et al., 2022; Ko et al., 2023). In contrast, our approach preserves knowledge without incurring such computational costs. Another related works, ResRep (Ding et al., 2021b) and SliceGPT (Ashkboos et al., 2024), also aim to preserve computational invariance. The knowledge transfer in HESSO-(CRIC) similarly seeks to maintain computational invariance but does so by focusing on preserving objective function levels. However, SliceGPT is restricted to the transformer architectures and requires manually injecting additional layers. ResRep is restricted to CNN architectures and require conducting structurally reparametrization via computing resetting gradients. HESSO-(CRIC) is architecture-agnostic, efficient and user-friendly, demonstrating both scalability and versatility.

**Neural Architecture Optimization.**   Another related realm is the optimization over pre-specified neural architecture. NAO (Luo et al., 2018) encodes the DNN architecture into a latent representation, search over the latent space, then decodes back to a revised architecture. NAT (Guo et al., 2019) performs operator transformation upon the given DNN to produce more accurate network. These approaches transform and improve the existing DNNs, yet not search an optimal sub-network. As a result, their produced networks are typically not significantly compact compared to the baseline models. Contrarily, our approach focuses on automatically and effectively discovering compact sub-networks given pre-specified DNNs via structured pruning.

## B  SUPPLEMENTARY FIGURES

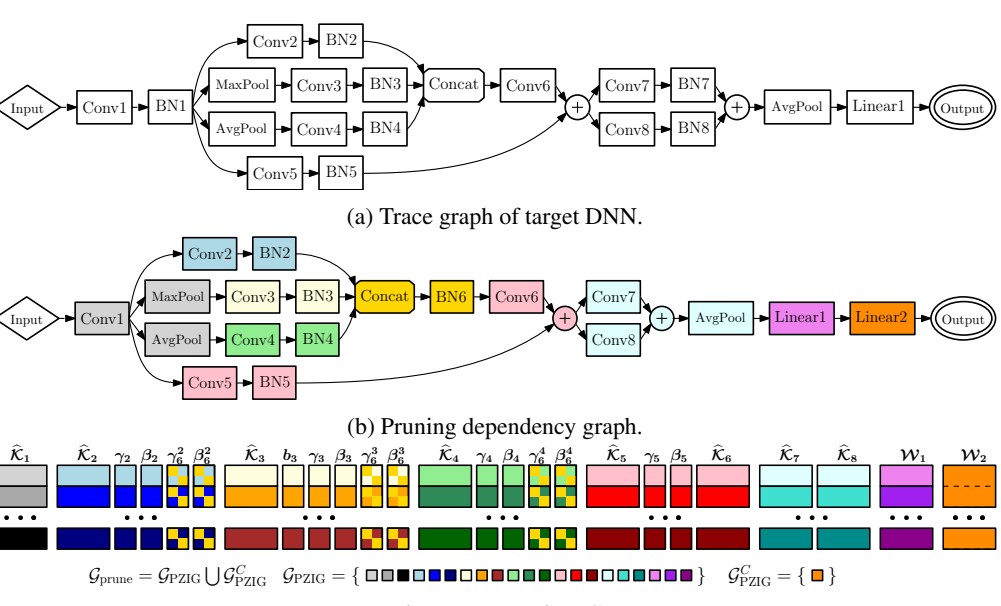

(a) Trace graph of target DNN.

(b) Pruning dependency graph.

(c) Pruning Zero-Invariant Groups.

Figure 4: Automated trainable variable partitions for one-shot structured pruning. Given the trace graph shown in Figure 4a, automatic pruning frameworks such as OTOv2 (Chen et al., 2023c) construct a pruning dependency graph shown as Figure 4b and partition the trainable variables as pruning zero-invariant groups $\mathcal{G}$ in Figure 4c.

## C  PROOF OF THEOREM 3.3

*Proof.* The statement is equivalent to that the violating cycle line 5-18 in Algorithm 2 terminates within finite number of steps. For convenience, we denote $\mathcal{V}_l$ as the violating set at $l$th cycle. The statement then becomes that there exists an $L < \infty$, such that $\mathcal{V}_L = \varnothing$. We now prove it as a two-step fashion.

At first, we show that the violating set at $l$th loop $\mathcal{V}_l$ is disjoint to those at all previous loops $\{\mathcal{V}_i\}_{i=0}^{i=l-1}$. This is true since the $\mathcal{V}_l$ is constructed excluding elements in the $l - 1$th historical set $\mathcal{H}_l$

$$\mathcal{V}_l \leftarrow \widehat{\mathcal{V}}_{l-1} \,/\, \mathcal{H}_{l-1}, \tag{6}$$

and $\mathcal{H}_{l-1}$ is the union of previous violating set $\mathcal{H}_{l-1} = \bigcup_{i=0}^{i=l-1} \mathcal{V}_i$. Therefore, $\mathcal{V}_l$ is disjoint to all violating sets $\{\mathcal{V}_i\}_{i=0}^{i=l-1}$.

Secondly, we prove on contraction. Suppose there exists no an $L < \infty$, such that $\mathcal{V}_L = \varnothing$. Since $\mathcal{V}_l$ is disjoint with $\{\mathcal{V}_i\}_{i=0}^{i=l-1}$, it implies that $\mathcal{V}_l$ must include previously unseen and new element from $\mathcal{G}$. Consequently, the historical set $H_l = \bigcup_{i=0}^{i=l} \mathcal{V}_i$ will have infinite number of elements as $l$ tends to $\infty$, *i.e.*,

$$\lim_{l \to \infty} |H_l| = \infty. \tag{7}$$

However, equation 7 contradicts that the historical set $H_l$ is a subset of group partition set $\mathcal{G}$, and the cardinality of $\mathcal{G}$ is finite. Therefore, we conclude the corrective redundancy identification cycle always terminates within a finite number of steps. $\square$

## D  SALIENCE SCORE

**Magnitude.** The importance of a parameter group can be determined by its magnitude. We further normalized against all the current important instances, mapping the score into the range $[0, 1]$. Heuristically, a group of variables with lower magnitude—implying they are closer to zero—typically contributes less to the model output. Therefore, such groups are often considered less important and more likely to be pruned.

$$\text{score}_{\text{mag}}([\boldsymbol{x}]_g) \leftarrow \|[\boldsymbol{x}]_g\|_2\,, \ \text{score}_{\text{mag}}([\boldsymbol{x}]_g) \leftarrow \text{score}_{\text{mag}}([\boldsymbol{x}]_g) / \sum_{g \in \mathcal{G}_I} \text{score}_{\text{mag}}([\boldsymbol{x}]_g). \tag{8}$$

**Average Magnitude.** While considering the overall magnitude can be useful, it may introduce bias by disproportionately favoring groups with more parameters, marking them as more important. To address this potential bias, the average magnitude is also considered. This metric measures the average parameter magnitude within each group, providing a normalized assessment that accounts for the number of parameters in each group. Consequently, the algorithm can more fairly compare groups of different sizes and prevent the overrepresentation of larger groups.

$$\text{score}_{\text{avg-mag}}([\boldsymbol{x}]_g) \leftarrow \|[\boldsymbol{x}]_g\|_2 \,/ \sqrt{|g|}\|, \ \text{score}_{\text{avg-mag}}([\boldsymbol{x}]_g) \leftarrow \text{score}_{\text{avg-mag}}([\boldsymbol{x}]_g) / \sum_{g \in \mathcal{G}_I} \text{score}_{\text{avg-mag}}([\boldsymbol{x}]_g).$$
$$\tag{9}$$

**Cosine Similarity.** Another criterion for determining group importance is the cosine similarity between the projection direction of parameter group and the negative gradient direction of the objective function. It can be calculated as the cosine similarity between $-[\boldsymbol{x}]_g$ and the negative gradient $-[\nabla f(\boldsymbol{x})]_g$, followed by a normalization to map onto a common scale. This metric evaluates whether projecting a group of parameters onto zero (*i.e.*, moving towards the origin along the negative parameter direction) aligns with a descent direction for the objective function. A descent direction is expected to decrease the objective function value, suggesting that pruning group of parameters onto zero may not significantly regress model's performance. As a result, such groups are more likely to be marked as redundant.

$$\text{score}_{\text{cosine}}([\boldsymbol{x}]_g, [\nabla f(\boldsymbol{x})]_g) \leftarrow [\boldsymbol{x}]_g^\top [\nabla f(\boldsymbol{x})]_g / (\|[\boldsymbol{x}]_g\| \|[\nabla f(\boldsymbol{x})]_g\|),$$
$$\text{score}_{\text{cosine}}([\boldsymbol{x}]_g, [\nabla f(\boldsymbol{x})]_g) \leftarrow \text{score}_{\text{cosine}}([\boldsymbol{x}]_g) / \sum_{g \in \mathcal{G}_I} \text{score}_{\text{cosine}}([\boldsymbol{x}]_g). \tag{10}$$

**Taylor Importance.** To further quantitatively approximate the effect of projecting the parameter group $[\boldsymbol{x}]_g$ onto zero on the objective function, we can employ the Taylor expansion. Taylor expansion could estimate the impact of small changes in the parameters on the function value, allowing us to consider varying orders of Taylor importance. In particular, the first-order Taylor expansion provides a linear approximation of the objective function around the current parameter point. The impact of setting $[\boldsymbol{x}]_g \to \boldsymbol{0}$ can be estimated by the dot product of the gradient and the change in parameters. It helps identify groups whose removal likely decrease objective function.

$$
\text{score}_{\text{Taylor-1st}}([\boldsymbol{x}]_g, [\nabla f(\boldsymbol{x})]_g) \leftarrow |f(\boldsymbol{x}) - f(\boldsymbol{x}|[\boldsymbol{x}]_g \to \boldsymbol{0})| \approx |[\boldsymbol{x}]_g^\top [\nabla f(\boldsymbol{x})]_g|,
$$
$$
\text{score}_{\text{Taylor-1st}}([\boldsymbol{x}]_g, [\nabla f(\boldsymbol{x})]_g) \leftarrow \text{score}_{\text{Taylor-1st}}([\boldsymbol{x}]_g, [\nabla f(\boldsymbol{x})]_g) / \sum_{g \in \mathcal{G}_I} \text{score}_{\text{Taylor-1st}}([\boldsymbol{x}]_g, [\nabla f(\boldsymbol{x})]_g).
$$
$$(11)$$

The second order Taylor importance is based on the second-order Taylor expansion. It includes the Hessian matrix, capturing the curvature of the objective function. This approximation considers not only the gradient but also the second derivative, providing a more accurate estimate of the impact of setting $[\boldsymbol{x}]_g \to \boldsymbol{0}$.

$$
\text{score}_{\text{Taylor-2nd}}([\boldsymbol{x}]_g, [\nabla f(\boldsymbol{x})]_g) \leftarrow |f(\boldsymbol{x}) - f(\boldsymbol{x}|[\boldsymbol{x}]_g \to \boldsymbol{0})| \approx [\boldsymbol{x}]_g^\top [\nabla f(\boldsymbol{x})]_g + \frac{1}{2}[\boldsymbol{x}]_g^\top [\nabla^2 f(\boldsymbol{x})]_g [\boldsymbol{x}]_g,
$$
$$
\text{score}_{\text{Taylor-2nd}}([\boldsymbol{x}]_g, [\nabla f(\boldsymbol{x})]_g) \leftarrow \text{score}_{\text{Taylor-2nd}}([\boldsymbol{x}]_g, [\nabla f(\boldsymbol{x})]_g) / \sum_{g \in \mathcal{G}_I} \text{score}_{\text{Taylor-2nd}}([\boldsymbol{x}]_g, [\nabla f(\boldsymbol{x})]_g).
$$
$$(12)$$

# E  COMPUTATIONAL COST ANALYSIS

In this section, we present the time and space complexities of HESSO-(CRIC).

Table 5: Notations.

| Symbol | Definition | Remark |
|---|---|---|
| $N$ | # of trainable variables with gradient | |
| $\mathcal{G}$ | The set of parameter groups | The common setup could be pruning/erasing zero-invariant groups. |
| $|\mathcal{G}|$ | The size of $G$ | **Typically negligible compared to $N$, see the below table.** |
| $T$ | # of training steps | |
| $T_{ht}$ | # of hybrid training steps | Set as $T_{ht} = T/10$ in our generic recipe. |
| $P$ | # of pruning periods | Set as $P = 10$ in our generic recipe. |
| $S$ | # of sampling steps in CRIC | Set as $S = 10$ in our generic recipe. |
| $C$ | # of cycles in CRIC | Empirically terminates within 10 cycles. |

Table 6: Magnitude Comparison Between $N$ and $\|G\|$.

| Model | $N$ | $|\mathcal{G}|$ | Ratio $|\mathcal{G}|/N$ |
|---|---|---|---|
| CARNx2 | $9.6 \times 10^5$ | $1.7 \times 10^3$ | $1.8 \times 10^{-3}$ |
| ResNet50 | $2.6 \times 10^7$ | $1.2 \times 10^4$ | $4.6 \times 10^{-4}$ |
| Yolov5-Large | $7.2 \times 10^6$ | $9.5 \times 10^3$ | $1.3 \times 10^{-3}$ |
| Bert-Base | $1.1 \times 10^8$ | $3.8 \times 10^4$ | $3.5 \times 10^{-4}$ |
| Phi2-2.7B | $2.7 \times 10^9$ | $4.1 \times 10^5$ | $1.5 \times 10^{-4}$ |

Table 7: Space and Time Complexity Comparison.

| Optimizer | Variant | Space Complexity (Peak) | Time Complexity | Space Complexity Projected onto Phi2 | Time Complexity Projected onto Phi2 |
|---|---|---|---|---|---|
| SGD | Standard | $O(2N)$ | $O(NT)$ | $O(2N)$ | $O(NT)$ |
| **HESSO** | **SGD** | $O(2N + \|G\|)$ | $O(NT + \|G\|P)$ | $O(2.00015N)$ | $O(NT + 1.5 \times 10^{-3}N)$ |
| **HESSO-CRIC** | **SGD** | $O(2N + \|G\|S)$ | $O(NT + \|G\|P + \|G\|SC)$ | $O(2.0015N)$ | $O(NT + 1.515 \times 10^{-1}N)$ |
| Adam/AdamW | Standard | $O(3N)$ | $O(2NT)$ | – | – |
| **HESSO** | **Adam/AdamW** | $O(3N + \|G\|)$ | $O(2NT + \|G\|P)$ | $O(3.00015N)$ | $O(2NT + 1.5 \times 10^{-3}N)$ |
| **HESSO-CRIC** | **Adam/AdamW** | $O(3N + \|G\|S)$ | $O(2NT + \|G\|P + \|G\|SC)$ | $O(3.0015N)$ | $O(2NT + 1.515 \times 10^{-1}N)$ |

HESSO-(CRIC) requires additional time and space complexities while the additions are negligible. In our numerous realistic applications besides the presented academic benchmarks, HESSO-(CRIC) are quite efficient, typically as efficient as standard training via vanilla optimizers.

# F   MORE EXPERIMENTAL RESULTS

## F.1   ABLATION STUDIES OF CRIC ON SALIENCY SCORES

The default format of CRIC primarily targets the most commonly used saliency scores that are sensitive to approximation errors caused by distances to the origin. For saliency scores with such higher sensitivities, CRIC's multiple sampling strategy—gathering information along the direction toward the origin—and its voting mechanism over historical statistics can effectively mitigate these identification issues.

To validate this, we have included a new ablation study for CRIC to demonstrate its improvements across varying saliency scores. As shown in the results, for commonly used saliency scores, CRIC effectively improves performance. However, magnitude and average magnitude benefits less from CRIC due to the persistence of large approximation errors, even as the groups of iterates move closer to the origin.

Table 8: Ablation Studies of CRIC on Zero-Shot Pruning Phi2.

| | Magnitude | | Avg Magnitude | | Cosine Similarity | | 1st Taylor | | 2nd Taylor | |
|---|---|---|---|---|---|---|---|---|---|---|
| | No CRIC | CRIC | No CRIC | CRIC | No CRIC | CRIC | No CRIC | CRIC | No CRIC | CRIC |
| Perplexity↓ | 629.1 | 489.4 | 713.5 | 644.6 | 525.5 | 53.4 | 438.3 | 28.6 | 378.2 | 37.1 |

Furthermore, for saliency scores whose approximation errors are not dependent on the distance to the origin, the philosophy of CRIC can still be applied with proper adaptations. In such cases, it is critical to analyze the root causes of the approximation errors for the given saliency scores. Based on these root causes, CRIC's multiple sampling strategy can be adjusted to collect more targeted signals, thereby reducing identification errors in these scenarios.

## F.2   COMPARATIVE ANALYSIS OF HYPER-PARAMETER TUNING EFFORTS

The key advantage of HESSO-(CRIC) over HSPGs in the OTO series lies in its white-box optimization design. Unlike HSPGs, which are black-box optimizers requiring extensive task-specific hyper-parameter tuning for optimal performance, HESSO-(CRIC) significantly reduces this sensitivity by design. To highlight this difference, we present a comparative analysis of the total number of training recipes required for three shared applications:

Table 9: Sparse optimization related hyper-parameter recipe comparisons.

| | HESSO-(CRIC) | DHSPG |
|---|---|---|
| Super-Resolution CARNx2 | General Recipe as described in Table 5 of manuscript. | Recipe #1: $\lambda = 10^{-2}$, $\lambda_{amplify} = 20$, $\epsilon = 0.0$, etc. |
| Image-Classification ResNet | General Recipe as described in Table 5 of manuscript. | Recipe #2: $\lambda = 10^{-3}$, $\lambda_{amplify} = 2$, $\epsilon = 0.95$, etc. |
| Question-Answering Bert | General Recipe as described in Table 5 of manuscript. | Recipe #3: $\lambda = 10^{-3}$, $\lambda_{amplify} = 2$, $\epsilon = 0.0$, etc. |
| **Total # of training recipes** | 1 | 3 |

As shown in the table, HESSO-(CRIC) achieves competitive or superior performance using a single general-purpose recipe, whereas DHSPG requires distinct task-specific hyper-parameter settings for each application.

Additionally, this comparison focuses only on hyper-parameters specific to sparse optimizers. Black-box optimizers like HSPGs inherently manage sparsity exploration processes, which demand further tuning of broader training parameters, such as learning rate schedules and the number of epochs. In contrast, the white-box design of HESSO-(CRIC) avoids such complexities, offering a more user-friendly, efficient, and practical solution.

## F.3   QUESTION AND ANSWERING

Later, we compare HESSO-(CRIC) with DHSPG, HSPG, and a representative proximal method ProxSSI (Deleu and Bengio, 2021) for pruning a transformer model Bert (Vaswani et al., 2017), evaluated on the SQuAD question-answering benchmark (Rajpurkar et al., 2016). It is important to note that proximal methods have been standard algorithms for solving sparse optimization problems for decades. However, they are not effective at exploring sparsity while maintaining model performance in deep learning applications (Dai et al., 2023).

As shown in Figure 5, HESSO, HESSO-CRIC, and DHSPG perform competitively on this task in terms of parameter reduction while maintaining F1 scores. However, DHSPG achieves these results after extensive hyper-parameter tuning, which is not convenient. HSPG penalizes all variables toward zero which severely restricts the optimization search space, leading to suboptimal performance. ProxSSI additionally lacks sufficient sparsity exploration capacity, being not comparable.

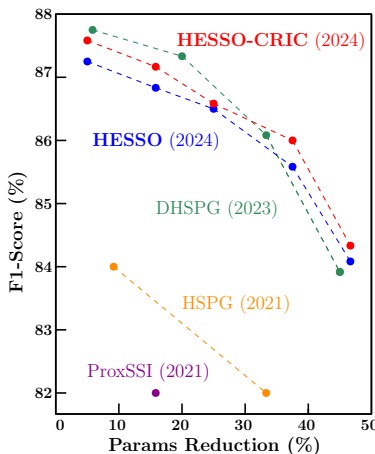

Figure 5: Bert on SQuAD.

## F.4 OBJECT DETECTION

Table 10: Structurally pruning Yolov5l on COCO.

| Method | # of Params | mAP$_{0.5}$ | mAP$_{0.5:0.95}$ |
|---|---|---|---|
| Baseline | 100% | 66.31% | 47.71% |
| HFP (Enderich et al., 2021) | 50% | 63.5% | 43.4% |
| TCFP (Jeon et al., 2022) | 50% | 61.8% | 42.7% |
| **HESSO** (30% group sparsity) | **49%** | **63.1%** | **44.4%** |
| **HESSO-CRIC** (30% group sparsity) | **49%** | **63.1%** | **44.5%** |

Next, we tested HESSO on the popular YOLO (Redmon et al., 2016) object detection model using the COCO benchmark dataset (Lin et al., 2014). Table 10 presents the structural pruning results for YOLOv5l (Jocher et al., 2022). Note that we selected YOLOv5l to facilitate comparisons with other existing benchmarks. We applied HESSO and HESSO-CRIC with a target group sparsity of 30%, resulting in a sub-network containing 49% of the original parameters. This allows for direct comparison with benchmarks that retain 50% of the model's parameters. The results show that a single run of HESSO and HESSO-CRIC achieved significantly higher Mean Average Precision (mAP) compared to other pruning approaches, which often require more complex, multi-stage procedures.

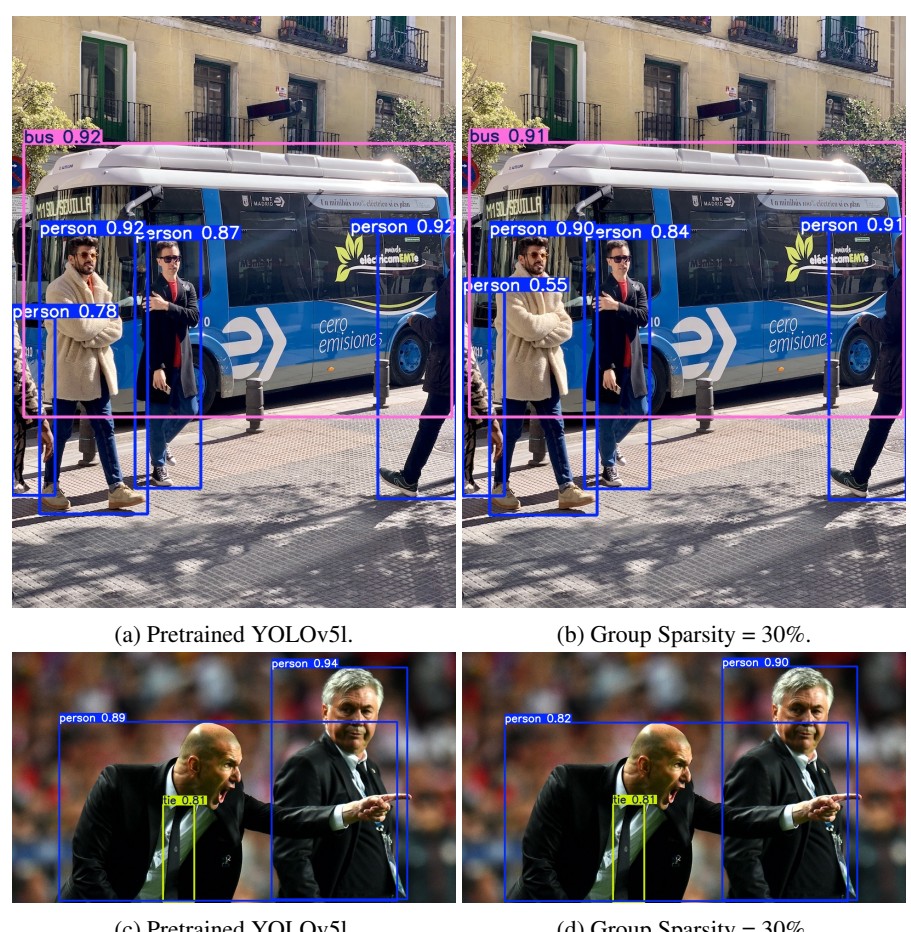

(a) Pretrained YOLOv5l.          (b) Group Sparsity = 30%.

(c) Pretrained YOLOv5l.          (d) Group Sparsity = 30%.

Figure 6: Visual examples of pruned YOLOv5l.

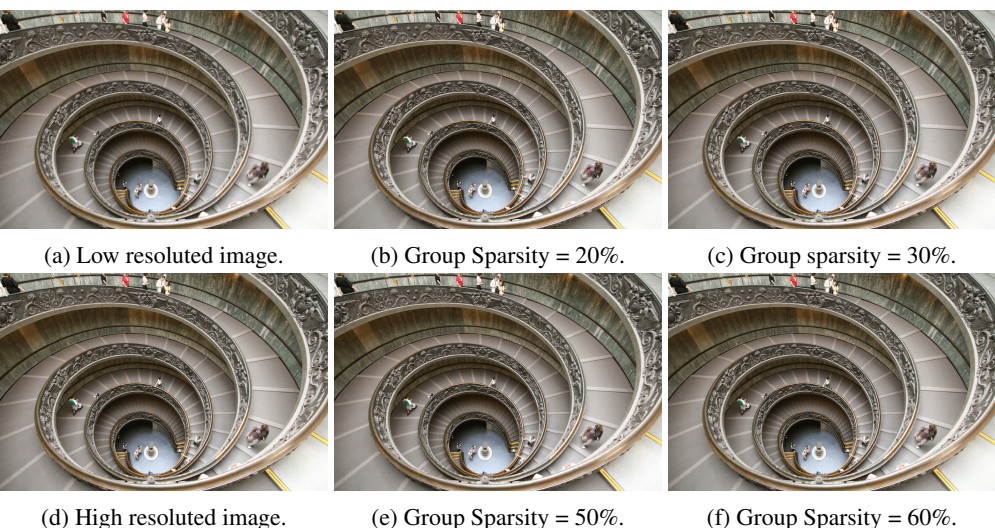

(a) Low resoluted image.    (b) Group Sparsity = 20%.    (c) Group sparsity = 30%.

(d) High resoluted image.    (e) Group Sparsity = 50%.    (f) Group Sparsity = 60%.

Figure 7: Visual examples of pruned CARNx2 produced HESSO-CRIC on Urban100.

Table 11: Structurally pruning Bert on SQuAD.

| Method | Group Sparsity | # of Params | F1-score |
|---|---|---|---|
| Baseline | 100% | 88.3% | |
| ProxSSI (Deleu and Bengio, 2021) | – | 83.4%[†] | 82.0% |
| HSPG (Chen et al., 2021b) | – | 91.0% | 84.1% |
| HSPG (Chen et al., 2021b) | – | 66.7% | 82.0% |
| DHSPG | 10% | 93.3% | 87.7% |
| DHSPG | 30% | 80.1% | 87.3% |
| DHSPG | 50% | 68.3% | 86.2% |
| DHSPG | 70% | 55.0% | 83.8% |
| **HESSO** | 10% | 94.78% | 87.20% |
| **HESSO** | 30% | 84.33% | 86.72% |
| **HESSO** | 50% | 73.88% | 86.46% |
| **HESSO** | 70% | 63.34% | 85.50% |
| **HESSO** | 90% | 53.0% | 84.25% |
| **HESSO-CRIC** | 10% | 94.78% | 87.48% |
| **HESSO-CRIC** | 30% | 84.32% | 87.10% |
| **HESSO-CRIC** | 50% | 73.88% | 86.50% |
| **HESSO-CRIC** | 70% | 63.44% | 85.96% |
| **HESSO-CRIC** | 90% | 53.0% | 84.10% |

[†] Approximate value based on (Deleu and Bengio, 2021).

