# OpenReview forum: "HESSO: Towards Automatic Efficient and User Friendly Any Neural Network Training and Pruning"
_ICLR.cc/2025/Conference — ICLR 2025 Conference Withdrawn Submission_

### Official Review · Reviewer_NVGL · 2024-10-30

**Soundness:** 2
**Presentation:** 3
**Contribution:** 1
**Rating:** 3
**Confidence:** 5

**Summary:**

This paper proposes HESSO, a novel optimizer designed for structured pruning of DNNs. HESSO aims to automatically and efficiently train any DNN to produce high-performing sub-networks with minimal need for hyper-parameter tuning, enhancing user-friendliness. Additionally, this paper introduces an extension of HESSO called HESSO-CRIC to address issues related to the irreversible performance collapse that sometimes occurs in pruned models. HESSO and HESSO-CRIC are evaluated across various applications, including computer vision and natural language processing, where it demonstrates competitive performance compared to existing baselines.

**Strengths:**

1. The paper is well-written, with clear organization and rigorous logic, making it understandable to readers.
2. HESSO can automatically and efficiently train and prune DNNs with any architecture, providing competitive performance, which holds broad significance in the field of neural network pruning.
3. The authors conducted a comprehensive analysis of how approximation errors of saliency score can lead to misidentification of indispensable structures in pruned DNNs, resulting in performance collapse. This part is well-organized, highly interesting, and very insightful.

**Weaknesses:**

1. I believe the main issue with this paper is the lack of many necessary comparison baselines, or the existing comparison baselines are outdated.
1) Section 4.3: In the CoFi [1] method, BERT-Base achieves a lossless F1 score on the SQuAD dataset with a 70% reduction in parameters (Figure 2 in CoFi). However, in the HESSO method, BERT-Base shows a drop of about 4% in F1 score with approximately a 50% reduction in parameters.
2) Section 4.4: HESSO seems to be inferior to the ATO [2] method, as ATO achieved a TOP-1 accuracy of 76.07% with a 61.7% reduction in FLOPs, only a -0.06% drop compared to the baseline. However, under the same FLOPs, the accuracy of HESSO is below 76%.
3) Section 4.6: The authors only compare with the SliceGPT method, lacking comparisons with some important structured pruning baselines for LLMs, such as LLM-Pruner (with LoRA fine-tuning) [3], LoRAPrune [4] and LoRAShear [5].
Due to the lack of the above baselines, I have concerns about the effectiveness of the HESSO method.
2. It seems that HESSO is simply improved version of OTO v3. As far as I know, OTO v3 also proposes a hybrid training scheme that applies different update mechanisms to important and redundant variable groups. HESSO does not seem to introduce any significant innovations but rather represents a minor technical improvement over OTO v3.
3. Different saliency scores likely result in varying approximation errors. The authors should demonstrate whether the CRIC mechanism effectively corrects each type of approximation error. Therefore, the authors should show the impact of different saliency scores in HESSO-CRIC on the final accuracy.
4. The phrase "can not be recovered given user resources" in Definition 3.1 is confusing, as it is evident that models with different parameters require significantly different training resources. The authors should clarify the definition of "user resources" more explicitly.

**Questions:**

1. In line 215, the authors state, “The selection of the saliency score in HESSO is flexible and can be tailored to different purposes.” I would like to know how the authors select the saliency score for different tasks in Sections 4.2-4.6.
2. In lines 503-507, the authors state, “Since both HESSO and HESSO-CRIC … without requiring tensor parallelism.” Does this mean that the HESSO method does not support models larger than 3 billion parameters? In addition, since both HESSO and HESSO-CRIC utilize full gradient information of LLMs, I would like to know the training cost of HESSO and HESSO-CRIC on LLMs.
[1] Structured Pruning Learns Compact and Accurate Models. ACL 2022 Main.
[2] Auto-Train-Once: Controller Network Guided Automatic Network Pruning
from Scratch. CVPR 2024.
[3] LLM-Pruner: On the Structural Pruning of Large Language Models. NeurIPS 2023.
[4] LoRAPrune: Structured Pruning Meets Low-Rank Parameter-Efficient Fine-Tuning. ACL 2024 Findings.
[5] LoRAShear: Efficient Large Language Model Structured Pruning and Knowledge Recovery. ArXiv.
[6] OTOv3: Towards Automatic Sub-Network Search Within General Super Deep Neural Networks. ArXiv.

---

> ### Author Response · Authors · 2024-11-22
> **Author responses (part 1)**
>
> We appreciate the reviewer NVGL for the constructive and insightful reviews. Please find our responses below to the questions. We hope that they could adequately address the questions and concerns.
>
> **Q1. Main issue is the lack of necessary comparison baselines.**
>
> A1. Thanks for the question. We have provided more comprehensive experiments and studies. Please refer to our general response (Part 1) and our responses to the below questions.
>
> **Q2. Section 4.3: In the CoFi [1] method, BERT-Base achieves a lossless F1 score on the SQuAD dataset with a 70% reduction in parameters (Figure 2 in CoFi). However, in the HESSO method, BERT-Base shows a drop of about 4% in F1 score with approximately a 50% reduction in parameters.**
>
> A2. Thanks for the insightful question. We found that the comparison between CoFi and HESSO-(CRIC) over Bert on SQuAD is not an apple-to-apple comparison. Comparing with HESSO-(CRIC), CoFi additionally conducts three complementary strategies.
>  - Exclude the heavy embedding layer from calculating the number of parameters. The embedding layer takes **22\% of total number of parameters** of Bert-base.
>  - Conduct layer-pruning to remove transformer layer entirely from the Bert.
>  - Layerwise distillation.
>
>  If we additionally include \#1 and \#2 complementary strategies, we could attain better performance, roughly 80% parameter reduction lossless compression.
>
>  In the revision, we have added CoFi to the related work section but decided not to include a direct quantitative comparison. This is primarily because automation of layer pruning is not there yet. For instance, in OTOv3, the erased DNN must adhere to ONNX formatting, which limits its broader usability.
>
> **Q3.  Section 4.4: HESSO seems to be inferior to the ATO [2] method, as ATO achieved a TOP-1 accuracy of 76.07% with a 61.7% reduction in FLOPs, only a -0.06% drop compared to the baseline. However, under the same FLOPs, the accuracy of HESSO is below 76%.**
>
>  A3. Thanks for the great question. Our answer is three-folds.
>  - **Hyperparameter discrepancy.**
>  The shared hyperparameters between ATO and HESSO-(CRIC) differ significantly, which could explain the observed accuracy gap. For example, ATO uses a batch size of 1024, while HESSO-(CRIC) uses 128. These differences in hyperparameter settings likely influence the performance outcomes. We are currently rerunning the experiments under identical setups to provide a more accurate comparison. However, due to a recent natural disaster in our local area, our servers are down, causing delays. We will include these updated results in a future revised version.
>
>  - **Tradeoff between simplicity & efficiency & generality versus performance.**
>  HESSO-(CRIC) and ATO, as subsequent developments of OTO-based approaches, prioritize different aspects:
>     - HESSO-(CRIC) emphasizes simplicity, efficiency, and generality while being tuning-free, which makes it broadly applicable and lightweight.
>     - ATO, on the other hand, focuses on optimizing performance, employing a bi-level optimization framework with a learnable mask and a ControlNet. While this enhances performance, it sacrifices simplicity, efficiency, and generality, making it more complex and resource-intensive.
>
> >    **Table. Comparison with ATO**
> >    ||HESSO-(CRIC)|ATO|DHSPG|
> >    |--|--|--|--|
> >    |Simplicty|$***$|$*$|$**$|
> >    |Efficiency|$***$|$*$|$**$|
> >    |Generality|$***$|$*$|$**$|
> >    |Tuning free|$***$|$*$|$**$|
> >    |Performance|$**(*)$|$***$|$**(*)$|
>
> - **Zero-shot NAS versus one-shot NAS.**
>  The comparison between HESSO-(CRIC) and ATO is analogous to the broader trend in NAS methodologies: zero-shot NAS versus one-shot NAS.
>     - One-shot NAS (e.g., ATO) employs bi-level differential optimization for higher performance but suffers from heavy search costs and complexity.
>     - Zero-shot NAS (e.g., HESSO-(CRIC)) prioritizes simplicity and efficiency by designing saliency scores for fast, high-performing architecture selection, with only a slight trade-off in performance.
>  This trend is evident in recent works, where zero-shot NAS has become the popular due to its practical advantages [1-5].
>
>  [1] Lee et al., AZ-NAS: Assembling Zero-Cost Proxies for Network Architecture Search. CVPR 2024
>
>  [2] Peng et al., SWAP-NAS: Sample-Wise Activation Patterns for Ultra-fast NAS. ICLR 2024
>
>  [3] Wei et al., Auto-prox: Training-free vision transformer architecture search via automatic proxy discovery. AAAI 2024
>
>  [4] Li et al., Zero-Shot Neural Architecture Search: Challenges, Solutions, and Opportunities. TPAMI 2024
>
>  [5] Jiang et al., MeCo: zero-shot NAS with one data and single forward pass via minimum eigenvalue of correlation. NeurIPS 2024
>
>  Based on the above, while HESSO-(CRIC) may not outperform ATO in terms of accuracy on ImageNet, its strengths in simplicity, efficiency, generality, and tuning-free optimization make it a significant contribution that is worthy a publication.

---

> ### Author Response · Authors · 2024-11-22
> **Author responses (part 2)**
>
> **Q4.  Lacking comparisons with some important structured pruning baselines for LLMs, such as LLM-Pruner (with LoRA fine-tuning), LoRAPrune and LoRAShear. Due to the lack of the above baselines, I have concerns about the effectiveness of the HESSO method.**
>
> A4. Thank you for the question. We agree that including more baseline comparisons can further demonstrate the effectiveness of HESSO. To address this, we applied LLM-Pruner, LoRA-Prune, and LoRAShear to the Phi-2 baselines. As detailed in our general responses (Part 1), HESSO still outperforms these methods.
>  We believe the primary reasons for this are:
>  - HESSO leverages full gradient information to compute saliency scores, enabling more precise pruning decisions.
>  - The CRIC mechanism enhances the stability of redundant structure identification.
>  - Full gradient utilization allows for more effective knowledge recovery in pruned models.
>
>  In contrast, LoRA-based PEFT methods are designed primarily for fine-tuning well-trained, large DNNs. These approaches are known to saturate quickly when applied to underfitting models, such as pruned LLMs, which often suffer significant performance degradation. This limitation highlights the advantage of HESSO’s full-gradient approach in maintaining performance.
>
> **Q5. OTO v3 also proposes a hybrid training scheme.**
>
>  A5. Thanks for raising this point. As outlined in our general response (Part 1), our contributions are twofolds: **HESSO** and **CRIC**. Our below answer will focus on the HESSO perspective.
>
>  **Hybrid training was proposed in OTOv2 and used in OTOv3 and LoraShear.** The concept of hybrid training is not first proposed in HESSO while was first introduced in OTOv2’s DHSPG. OTOv3 further refined this concept and introduced H2SPG, which incorporates a hierarchical search to ensure validity of sub-networks after layer pruning. Similarly, LoRAShear’s LHSPG employs a hybrid training strategy but serves as a LoRA variant.
>
>  **Existing hybrid training are black-box and hard-to-use.**  However, the hybrid training schemas in DHSPG, H2SPG, and LHSPG are all black-box optimizers, which have significant limitations. Their success depends heavily on: significant hyper-parameter tuning efforts and substaintial expertise in sparse optimization. Consequentially, they are still not user-friendly and generic in practice.
>
>  **White-box structured-pruning-aware training optimizer.** To address these limitations, we introduce HESSO, a simple and effective method. HESSO is the first white-box structured-pruning-aware training optimizer to the best of our knowledge. HESSO is impressivelly simple, generic, almost tuning-free, user-friendly, and high-performing. These advantages bring significant practical value and distinguish HESSO from previous methods, making it a meaningful contribution to the field.
>
> **Q6. CRIC on correcting approximation error of varying saliency scores**
>
> A6. It is a great suggestion. The current format of CRIC mainly targets at the perhaps most popular saliency scores which are sensitive to the approximation errors caused by distances to the origin. For these saliency scores with such higher sensitivities, CIRC’s multiple sampling strategy, i.e., gathering information along the direction toward the origin, and its voting mechanism over the historical statistics can effectively mitigate these identification issues.
>
>  To validate it, we have included a new ablation study for CRIC to demonstrate its improvement over varying saliency scores. We can see that for the commonly used saliency scores, CRIC could effectively correct their performance. However, the Magnitude receives less benefits due to the persistence of large approximation errors even when the iterate is closer to the origin.
>
> > **Table. Ablation Studies of CRIC on Zero-Shot Pruning Phi2.**
> > ||Magnitude|--|1st Taylor|--|2nd Taylor|--|
> > |--|--|--|--|--|--|--|
> > ||No CRIC|CRIC|No CRIC|CRIC|No CRIC|CRIC|
> > |Perplexity↓|629.1|489.4|438.3|28.6|378.2|37.1|
>
> **Q7.The phrase "can not be recovered given user resources" in Definition 3.1 is ambiguous.**
>
> A7. That is a great point. We have revised the definition as follows with higher clarity.
>
>  **Definition (Indispensable Structure).** Given a deep neural network $\mathcal{M}$, a minimally removal structure is called indispensable if removing it from $\mathcal{M}$  would cause significant performance  degradation, which can not be recovered given user resources. In particular, we say a minimally removal structure as $\epsilon$-indispensable associated with an objective $f$ if pruning the variables $[x]_g\to {0}$ deteriorates $f$ at least $\epsilon$, ie, $f(x| [x]_g\to {0})\geq f({x})+\epsilon$ for a minimization optimization problem. The degradation $\epsilon$ can not be recovered by (i) keeping training $\mathcal{M}$, (ii) the  training cost such as GPU days exceeding user budget, or (iii) the training receipt for $\mathcal{M}$ is black-box and hard to be reproduced.

---

> ### Author Response · Authors · 2024-11-22
> **Author responses (part 3)**
>
> **Q8.  In line 215, the authors state, “The selection of the saliency score in HESSO is flexible and can be tailored to different purposes.” I would like to know how the authors select the saliency score for different tasks in Sections 4.2-4.6.**
>
>  A8. For all the experiments in the main text, we used a shared recipe that equally consider each saliency score in Appendix D.
>
> **Q9.  In lines 503-507, the authors state, “Since both HESSO and HESSO-CRIC … without requiring tensor parallelism.” Does this mean that the HESSO method does not support models larger than 3 billion parameters? In addition, since both HESSO and HESSO-CRIC utilize full gradient information of LLMs, I would like to know the training cost of HESSO and HESSO-CRIC on LLMs.**
>
> A9. Thanks for the great question. To clarify, HESSO-(CRIC) can indeed be applied to DNNs with more than 3 billion parameters. Specifically, the largest supported model size, without considering tensor parallelism, is determined by whether the training can be performed on a single GPU.
>
> As outlined in our general response (Part 2), HESSO-(CRIC) shares almost the same space complexity as standard optimizers. Therefore, as long as the model can be trained using a standard optimizer on a single GPU, HESSO-(CRIC) can also handle it.
>
> For resource-constrained scenarios, we can adapt HESSO to a more parameter-efficient variant, such as HESSO-LoRA, to better suit these situations. Such variant could enable efficient training of larger models even with limited resources.
>
> Yours,
>
> Authors

---

> > ### Author Response · Authors · 2024-12-02
> >
> > We would like to thank reviewer NVGL once again for the time and effort in reviewing our work.
> >
> > We hope our response helped address your concerns regarding more numerical comparisons and other questions. As the discussion period ended today, we would appreciate it if you could let us know whether our responses were satisfactory and share any further thoughts you might have.
> >
> > Please do not hesitate to let us know if any further question or concern.

---

### Official Review · Reviewer_SYtw · 2024-10-31

**Soundness:** 3
**Presentation:** 3
**Contribution:** 3
**Rating:** 6
**Confidence:** 3

**Summary:**

The manuscript presents a novel approach to deep neural network (DNN) training and structured pruning through Hybrid Efficient Structured Sparse Optimize (HESSO). The topic is highly relevant in the current AI landscape, where deploying efficient models in resource-constrained environments is critical. HESSO stands out by simplifying hyper-parameter setup and offering a progressive pruning strategy that enhances user-friendliness. In addition, the authors also discuss the limitations of the proposed method and the approximation errors of salience scores.

**Strengths:**

1. This paper is well-motivated, and the topic is highly relevant in the current AI landscape, where deploying efficient models in resource-constrained environments is critical.
2. The manuscript provides a well-structured theoretical framework for HESSO, which helps justify the design choices made in the method.
3. The authors conduct extensive experiments in an array of tasks.

**Weaknesses:**

1. While HESSO aims to simplify hyper-parameter tuning, the manuscript does not provide sufficient evidence to support the claim of reduced sensitivity. A comparative analysis of tuning efforts required for HESSO versus traditional methods is needed.
2. The scalability of HESSO to larger models is not well demonstrated. A table or figure listing the computational cost of applying HESSO to models across different model sizes should be provided.

**Questions:**

1. How does HESSO manage the trade-off between sparsity and performance? Is there a recommended range for optimal sparsity levels?
2. Is there a risk that the method could lead to overfitting, especially during the pruning phase? Maybe evaluation in held-out datasets are needed.

---

> ### Author Response · Authors · 2024-11-22
> **Author responses**
>
> We appreciate the reviewer SYtw for the constructive and insightful reviews. Please find our responses below to the questions. We hope that they could adequotely address the questions and concerns.
>
> **Q1. Need comparative analysis of tuning efforts for HESSO versus traditional methods is needed.**
>
> A1. Thanks for the great suggestion. The hyper-parameter tuning advantages from HESSO-(CRIC) against HSPGs in OTO series largely stem from the white-box optimization design. Unlike HSPGs, which are black-box optimizers and require extensive case-by-case hyper-parameter tuning to achieve optimal performance, HESSO-(CRIC) is designed to minimize such sensitivity.
>
> To provide the comparitive analysis, we at first provide the total number of training receips to three shared applications.
>
> > **Table. Sparse optimization related hyper-parameter recipe comparisons**
> > ||HESSO-(CRIC)|DHSPG|
> > |--|--|--|
> > |Super-Resolution CARNx2|General Recipe as described in Table 1 of manuscript.|Recipe #1: $\lambda=10^{-2}$, $\lambda_{amplify}=20$, $\epsilon=0.0$, etc.|
> > |Image-Classification ResNet|General Recipe as described in Table 1 of manuscript.|Recipe #2: $\lambda=10^{-3}$, $\lambda_{amplify}=2$, $\epsilon=0.95$, etc.|
> > |Question-Answering Bert|General Recipe as described in Table 1 of manuscript.|Recipe #3: $\lambda=10^{-3}$, $\lambda_{amplify}=2$, $\epsilon=0.0$, etc.|
> > |Total \# of training recipes|**1**|3|
>
> The table above demonstrates that HESSO-(CRIC) requires only a single recipe to achieve competitive or even superior performance compared to state-of-the-art methods. In contrast, DHSPG necessitates task-specific hyper-parameter tuning to optimize its performance.
>
> Moreover, the table focuses solely on the hyper-parameters specific to sparse optimizers. Black-box optimizers like HSPGs implicitly control the sparsity exploration process, which requires additional tuning of the overall training pipeline (e.g., learning rate schedules, number of epochs). In contrast, white-box optimizers such as HESSO-(CRIC) eliminate this inconvenience, offering substantial practical benefits in terms of simplicity and efficiency.
>
> Finally, in the revision, we also highlighted that the sensitivities of HSPGs related optimizers have been shown in previous literatures such as ATO and AdaHSPG+ (line 77-78).
>
> **Q2.  The scalability of HESSO to larger models is not well demonstrated. A table or figure listing the computational cost of applying HESSO to models across different model sizes should be provided.**
>
> A2. Thanks for the question regarding the computational cost. Please refer to our general responses (Part 2) for detailed information. Based on our experience, the computational cost of HESSO-CRIC is on the par to that of standard optimizers.
>
> **Q3. How does HESSO manage the trade-off between sparsity and performance? Is there a recommended range for optimal sparsity levels?**
>
> A3. Thanks for the question. Based on our extensive empirical experience, including scenarios beyond the academic benchmarks presented in the paper, the optimal sparsity levels largely depend on factors such as the transparency of the training pipeline (e.g., datasets and training recipes) and the available user resources, such as GPU capacity.
>
> - **Under full transparency of training pipeline and sufficient resources.**
>     Our experience across domains indicates that full DNNs can typically reduce their model size by 60–80\% while maintaining competitive performance. In some cases, pruning more mildly (e.g., a 20–30\% model size reduction) can even lead to slight performance improvements. The size of pruned sub-network is generally quadratically proportional to (1 - sparsity level), assuming all parameters in the target DNN are prunable.
>
> - **Under limited transparency or user-resources.**
>    In scenarios with restricted resources or partial knowledge of the training pipeline, the optimal sparsity level varies significantly on a case-by-case basis. For instance,  for some LLMs, like Llamav1 and v2, pruning 20% with limited resource does not affect the performance that much. However, for Phi-2, pruning 20\% under limited resource might degrade the performance more severely. For some computer vision based model, using limited dataset can still largely recover the performance a pruned model with 60\% parameter reduction.
>
>
> **Q4. Is there a risk that the method could lead to overfitting, especially during the pruning phase? Maybe evaluation in held-out datasets are needed.**
>
> A4. That is a great point. For sake of seamless integration as standard optimizer, HESSO-(CRIC) currently relies on using training dataset to compute the saliency scores. Such strategy generally works well, while indeed may result in some overfitting in specific situations. To pursure the best performance on validation set, having a held-out dataset to compute saliency score is a great idea and should indeed help in this situation.
>
>
> Yours,
>
> Authors

---

> > ### Comment · Reviewer_SYtw · 2024-11-27
> >
> > Thank you for your detailed responses. However, my concerns remain. The additional responses about sparsity trade-offs and overfitting, while informative, further highlight the method's limitations in terms of consistency and reliability across different scenarios.

---

> ### Author Response · Authors · 2024-11-27
>
> We appreciate the reviewer for the further responses. Please find our below responses.
>
> **Q5. How to avoid overfitting?**
>
> A5. Thanks for the great question.
>
> We think that in the most cases, using training set to compute saliency scores is good enough to achieve competitive even better performance to the state-of-the-arts as presented in the main text. In some remaining cases to pursue further advances, we agree that leveraging held-out datasets could provide further gains. We can provide the flexibility to support both demands.
>
> In particular, during the hybrid training phase, we can leverage held-out dataset to compute the saliency score rather than fully relying on the training set. It could effectively address potential over-fitting issue. The training workflow would require some adjustments accordingly. Due to the limited time left, we might not be able to provide ablation studies with this regard in time, but will incorporate this insightful suggestion into our next revision.
>
> **Q6. Optimal sparsity-levels vary across different scenarios.**
>
> A6. That is a insightful question. The main factor for the optimal sparsity level is the transparency of training recipes and resources to reproduce the baseline training upon our experience.
>
> - **Reliable optimal sparsity level for full transparent scenarios.** We found that given sufficient resources and full transparency, the claimed 60\%-80\% model reduction with competitive performance is reliable across different applications under HESSO. However, for the cases of limited resources and black-box training recipes, the optimal sparsity levels indeed vary case-by-case. We will revise our manuscript to properly reflect this perspective and highlight that our discussed tuning-free advantages of HESSO largely refers to the former case.
>
> - **Reliable solution for finding optimal sparsity level for limited resources case.** For the latter limited-resources case, injecting some early stop during the iterative pruning of HESSO could be a simple and effective solution to discover the optimal pruning ratio. In details, we can start with a sufficient largely target sparsity level and pruning periods, during each pruning period, only a small portion of redundant groups are incrementally identified.  We could monitor the performance evolution during the incremental redundancy identification, proceed early stop when the performance is significantly dropped, and return the stopped sparsity level as the approximately optimal ratio.

---

### Official Review · Reviewer_qBt9 · 2024-11-04

**Soundness:** 1
**Presentation:** 2
**Contribution:** 2
**Rating:** 3
**Confidence:** 4

**Summary:**

The paper addresses the challenge of deploying large deep neural networks by proposing **HESSO**, a new optimizer that enables automatic one-shot joint training and structured pruning for various DNN architectures and tasks. HESSO simplifies the hyper-parameter setup and employs a progressive pruning strategy to explicitly control sparsity exploration, enhancing usability. It optionally integrates a Corrective Redundancy Identification Cycle mechanism to accurately identify redundant groups, minimizing the risk of irreversible performance degradation caused by pruning indispensable structures.  The authors have conducted experiments on various type of tasks and models.

**Strengths:**

1.By simplifying hyper-parameter tuning and explicitly controlling sparsity exploration, HESSO enhances usability, making it accessible to practitioners without deep optimization expertise.

2. The optimizer is designed to be applicable to any DNN architecture and task, increasing its potential impact across various domains and applications.

**Weaknesses:**

1. As a paper on structured pruning, it does not compare with state-of-the-art methods on ImageNet, such as OFA, EagleEye ResRep, etc. The comparisons are too simplistic, making it impossible to see the advantages of the proposed method. Furthermore, in experiments on large models, it only compares with SliceGPT, which is not tailored for structured pruning. More mainstream baseline methods like LLMPruner need to be included for comparison. Overall, the experimental section significantly dampens my enthusiasm for this paper.

2. The authors' discussion of related work in the field is rather lacking and not sufficiently grounded. The entire introduction, methodology, and experimental discussions focus only on OTO-based papers[4,5,6]. However, as far as I know, there are many papers on automatic structure search for structured pruning, including OFA[1], EagleEye[2], etc. I suggest that the authors provide a more comprehensive coverage of the field, which would effectively enhance the quality of this paper.

[1] Cai, H., Gan, C., Wang, T., Zhang, Z. and Han, S., 2019. Once-for-all: Train one network and specialize it for efficient deployment. In ICLR, 2020.

[2] Li, B., Wu, B., Su, J. and Wang, G., 2020. Eagleeye: Fast sub-net evaluation for efficient neural network pruning. In ECCV, 2020.

[3] Ding, X., Hao, T., Tan, J., Liu, J., Han, J., Guo, Y. and Ding, G., 2021. Resrep: Lossless cnn pruning via decoupling remembering and forgetting. In ICCV, 2021.

[4] Chen, T., Ji, B., Ding, T., Fang, B., Wang, G., Zhu, Z., Liang, L., Shi, Y., Yi, S. and Tu, X., 2021. Only train once: A one-shot neural network training and pruning framework. In NeurIPs, 2021.

[5] Chen, T., Liang, L., Ding, T., Zhu, Z. and Zharkov, I., 2023. Otov2: Automatic, generic, user-friendly. In Arxiv, 2022.

[6] Chen, T., Ding, T., Zhu, Z., Chen, Z., Wu, H., Zharkov, I. and Liang, L., 2023. OTOv3: Automatic Architecture-Agnostic Neural Network Training and Compression from Structured Pruning to Erasing Operators. In Arxiv, 2023.

**Questions:**

Please see the weakness part.

---

> ### Author Response · Authors · 2024-11-22
> **Author responses**
>
> We appreciate the reviewer qBt9 for the constructive and insightful reviews. Please find our responses below to the questions. We hope that they could adequotely address the questions and concerns.
>
> **Q1. More comparison on ImageNet and LLMs.**
>
> A1.Thanks for the suggestion. We agree that adding more comparisons in these areas could significantly enhance the quality of this work. In response, we have conducted additional experiments, as detailed in our general responses (Part 1).
>
> Specifically, in the MobileNet search space, we started from an OFA_{LARGE} model, which achieves 80% accuracy on ImageNet. We then applied HESSO-(CRIC) to discover pruned sub-networks of equivalent sizes that demonstrated higher accuracy than other OFA counterparts.
>
> For large language models (LLMs), we added comparisons with LLM-Pruner, LoRA-Prune, and LoRAShear. Our initial choice to include SliceGPT stems from its outstanding performance preservation than many prominent methods. Meanwhile, LLM-Pruner, LoRA-Prune, and LoRAShear are LoRA-based approaches, making direct comparisons with our full-gradient-based pruning-aware-training algorithms less fair. This is because LoRA is typically used for fine-tuning well-trained DNNs and has been shown to be less effective at gathering knowledge for underfitted models, whereas our approach is designed to address broader scenarios.
>
> **Q2. Expand related work and compare with OFA, EagleEye, and ResRep.**
>
> A2. Thank you for the suggestion. We have expanded our related works and discussions in the main text to include comparisons with OFA, Eagle Eye, and ResRep.
>
> -  **OFA** automates the generation of sub-networks for all target hardwares within a pre-defined search space. In contrast, OTO based methods focus on automatically and structurally pruning any DNN to produce high-performing sub-networks without the need for a pre-defined search space. (Section 2)
>
> - **EagleEye** utilizes adaptive batch normalization to improve the evaluation and selection of a fixed pruning strategy. On the other hand, we employ CRIC, an iterative voting mechanism that leverages saliency scores to identify redundant groups. (Section 2)
>
> - **ResRep** uses structural re-parameterization to achieve computational invariance in CNNs, similar to SliceGPT's for transformers. These methods, however, are architecture-specific and require additional engineering efforts. Our approach employs a flexible hybrid training method, enabling architecture-agnostic computational invariance in a white-box manner. We have also compared our method with ResRep on ResNet-50 using ImageNet. (Section 3.2 and Appendix A)
>
> Yours,
>
> Authors

---

> > ### Author Response · Authors · 2024-12-02
> >
> > We would like to thank reviewer qBt9 once again for the time and effort in reviewing our work.
> >
> > We hope our response helped address your concerns regarding additional related works and comparisons. As the discussion period ended today, we would appreciate it if you could let us know whether our responses were satisfactory and share any further thoughts you might have.
> >
> > Please do not hesitate to let us know if any further question or concern.

---

### Official Review · Reviewer_oZCz · 2024-11-04

**Soundness:** 3
**Presentation:** 3
**Contribution:** 3
**Rating:** 6
**Confidence:** 3

**Summary:**

This paper proposes Hybrid Efficient Structured Sparse Optimizer (HESSO), a structured pruning algorithm. In addition, for reliably identifying the indispensable structure, the authors introduce Corrective Redundant Identification Cycle (CRIC). The experimental results show that HESSO with CRIC plugged in can perform well on various types of models, e.g., image classification, object detection and large language models.

**Strengths:**

1. This paper is well written and organized.
2. This paper provides thorough experiments, by evaluating the performance of HESSO on various tasks and comparing with many other pruning methods.
3. It is demonstrated that HESSO is able to achieve strong and impressive results with a considerable pruning ratio.

**Weaknesses:**

See questions.

**Questions:**

1. What is the computational cost of employing HESSO? I am wondering whether it will significantly increase the computational memory and flow training speed.
2. Seems CIRC can be used as a generic module to identify the indispensable structures. Can it be applied to other pruning algorithms and what is the performance?

---

> ### Comment · Reviewer_oZCz · 2024-11-19
>
> After reading the comments from reviewer qBt9 and NVGL, I do agree that some necessary comparisons with baselines are missing from the manuscript. Therefore, I lower my score to 5 and hope the authors can proofread according to the comments.

---

> > ### Author Response · Authors · 2024-11-19
> >
> > We thank the reviewer oZCz for the careful consideration, supports, and suggestions to incorporate the valued comments.
> >
> > To share some quick update, we are closed to complete all requested experiments and will provide comprehensive responses shortly in this week.
> >
> > Look forward to further discussion then.

---

> ### Author Response · Authors · 2024-11-22
> **Author responses**
>
> We appreciate the reviewer oZCz for the constructive and insightful reviews. Please find our below individual and above general responses. We hope that they could adequately address the questions and concerns.
>
> **Q1. What is the computational cost of employing HESSO?**
>
> A1. Thanks for the question. Please refer to our general responses (Part 2). Upon the experience so far, the computational cost of HESSO-CRIC is on the par to the standard optimizers.
>
> **Q2. Seems CRIC can be used as a generic module to identify the indispensable structures. Can it be applied to other pruning algorithms and what is the performance?**
>
> A2. Thanks for the insightful question. You are right that CRIC could serve as a generic module that can be integrated into saliency-score-based pruning algorithms to provide more reliable identification of redundant structures.
>
> The default format of CRIC mainly targets at the perhaps most popular saliency scores which are sensitive to the approximation errors caused by distances to the origin. For these saliency scores with such higher sensitivities, CIRC’s multiple sampling strategy, i.e., gathering information along the direction toward the origin, and its voting mechanism over the historical statistics can effectively mitigate these identification issues.
>
> To validate it, we have included a new ablation study for CRIC to demonstrate its improvement over varying saliency scores. We can see that for the commonly used saliency scores, CRIC could effectively correct their performance. However, the Magnitude receives less benefits due to the persistence of large approximation errors even when the iterate is closer to the origin.
>
> > **Table. Ablation Studies of CRIC on Zero-Shot Pruning Phi2.**
> > ||Magnitude|--|1st Taylor|--|2nd Taylor|--|
> > |--|--|--|--|--|--|--|
> > ||No CRIC|CRIC|No CRIC|CRIC|No CRIC|CRIC|
> > |Perplexity↓|629.1|489.4|438.3|28.6|378.2|37.1|
>
> Furthermore, for saliency scores whose approximation errors are not dependent on the distance to the origin, the philosophy of CIRC can still be applied with proper adaptations. In such cases, it is crucial to analyze the root causes of the approximation errors for the given saliency scores. Based on the root cause, CIRC's multiple sampling strategy can be adjusted to collect more targeted signals, helping reduce identification errors in these scenarios as well.
>
> **Q3. Add more comparisons and baselines.**
>
> A3. Thanks for the suggestion. We agree that incorporating additional comparisons would significantly strengthen this work. In response, we have conducted more experiments, which are outlined in our general responses (Part 1).
>
> For the MobileNet search space, we started with an OFA_{LARGE} model, achieving 80% accuracy on ImageNet. Using HESSO-(CRIC), we identified pruned sub-networks of equivalent sizes but with improved accuracy compared to other OFA counterparts.
>
> In the context of large language models (LLMs), we added comparisons with LLM-Pruner, LoRA-Prune, and LoRAShear. In our initial version, SliceGPT was included for comparison due to its excellent performance. Meanwhile, it is worth noting that LLM-Pruner, LoRA-Prune, and LoRAShear are LoRA-based techniques. Lora primarily focuses on fine-tuning well-trained models. This makes direct comparisons with our full-gradient-based approach less fair for these approaches, as LoRA is generally less effective in capturing knowledge for underfitted models—a scenario our approach is well-equipped to handle.
>
>
> Yours,
>
> Authors

---

> > ### Comment · Reviewer_oZCz · 2024-11-28
> >
> > Thank the authors for addressing my concerns. I increase my score to 6.

---

> > > ### Author Response · Authors · 2024-11-28
> > >
> > > We appreciate the reviewer for increasing the score.

---

### Author Response · Authors · 2024-11-13
**Thank all reviewers for their careful reviews and insightful feedbacks.**

Dear reviewers and ACs,

We greatly thank all the reviewers for the careful review and for providing insightful and constructive feedbacks. We believe that incorporating these valuable suggestions could significantly improve the quality of our paper.  We are also happy to see that the significance and practical impacts of the work, writing quality, structural logics, experiment broadness, and theoretical boundness have been positively recognized by the reviewers.

At present, we are working on revising the manuscript and will tackle all raised concerns and questions within this discussion period, such as including more recent baselines and more comparisons.

Look forward to our further discussion.

Thanks all again!

Best,

Authors of Paper 3237

---

### Author Response · Authors · 2024-11-19
**General response (part 1)**

We greatly thank all reviewers for their insightful feedback and constructive suggestions, which have significantly improved the quality of our work. Below, we summarize the main contributions of this work, explain the intent behind our initial experimental designs, and address the shared concern regarding the inclusion of more baselines and comparisons.

### **Contributions of this work**

The contributions are two folds.
 - **HESSO (the first white-box structured-pruning-aware training optimizer)**: We introduce HESSO, the first white-box structured-pruning-aware training optimizer designed for generic applications and deep neural networks (DNNs) to our knowledge. HESSO is simple and effective. In comparison to prior works, such as black-box optimizers like HSPGs or bi-level optimization schemes like ATO, HESSO offers several key advantages:
     - **Hyperparameter tuning-free**: We provide a general pruning aware training recipe, used across all of our experiments. It requires no expertise in sparse optimization, making it easy to apply across a wide range of applications.
     - **Broad applicability**: Its generic nature allows it to be used in diverse applications and DNNs, demonstrating its utility for practical use cases.
- **CRIC (Corrective Redundancy Identification Cycle)**: Current popular saliency-score-based pruning algorithms sometimes fail to preserve the performance of pruned DNNs due to the existence of indispensable structures. We identify the root cause as the approximation error inherent in existing saliency scores. We propose CRIC, a novel method to reliably identify redundant structures even for the DNNs wherein indispensable structures exist. CRIC comes with theoretical guarantees regarding its efficiency and accuracy, ensuring robust pruning results.

### **Experimental design intent: Broadness and Representativeness**

Our goal was to develop a generic white-box pruning-aware training optimizer. To validate its broad applicability, we designed experiments covering five distinct applications, i.e., super-resolution, image-classification, object detection, question and answering, and autoregressive LLM. For each application, we selected challenging counterparts to increase representativeness. Many of these selected applications have been used as main benchmarks in prior domain-specific publications, further ensuring the rigor and relevance of our evaluation.

### **Main issues raised during reviews: increasing the Depth of specific experiments.**

We agree with the reviewers that delving deeper into ImageNet and large language models (LLMs) would better align our work with standard pruning literature. To address this, we conducted additional experiments:
- **ImageNet**: We included comparisons with suggested benchmarks to showcase HESSO's performance more comprehensively.
- **LLMs**: We expanded comparisons to include LLM-Pruner, LoraPrune, and LoraShear, which used Lora either in pruning or fine-tuning. The significant margin of HESSO-CRIC largely comes from the CRIC and the full gradient information to more adequately learn and transfer knowledge. However, Lora has been shown easy to be saturated to the underfit DNNs, e.g., pruned DNNs.

> |Method|Grad Type|Iterative Pruning|Hybrid Training|Dynamic Recover|
>|--|--|--|--|--|
> |LLM-Pruner|Full grad for pruning and Lora for fine-tuning|❌|❌|❌|
> |LoraPrune|Lora for pruning and fine-tuning|✅|❌|❌|
> |LoraShear|Lora for pruning and fine-tuning|✅|✅|✅|
> |**HESSO-CRIC**|**Full grad for pruning-aware training**|✅|✅|--|

> **Table 1 Structurally Prune Phi2-2.7B**
> |Pruning Ratio|Method|Avg Perf|Pruning Ratio|Method|Avg Perf|Pruning Ratio|Method|Avg Perf|
> |--|--|--|--|--|--|--|--|--|
> |20%|SliceGPT|60.22|25%|SliceGPT|57.11|30%|SliceGPT|44.53|
> ||LLM-Pruner|46.75||LLM-Pruner|45.00||LLM-Pruner|44.17|
> ||LoraPrune|50.83||LoraPrune|48.83||LoraPrune|46.92|
> ||LoraShear|51.81||LoraShear|48.95||LoraShear|47.45|
> ||**HESSO-CRIC**|**60.67**||**HESSO-CRIC**|**58.74**||**HESSO-CRIC**|**55.62**|


> **Table 2 Structurally Prune over MobileNet Super-Network**
> |Method|Num of Params (M)|MACs (M)|Top Acc-1 (%)|
> |--|--|--|--|
> |MobileNetV2|3.4|300|72.0|
> |MobileNetv3-Large|5.4|219|75.2|
> |OFA # 75|5.81|230|76.9|
> |EagleEye|--|~284| 70.9|
> |**HESSO**|5.60|220|78.2|
> |**HESSO-CRIC**|5.71|225|**78.6**|

We will provide the revised manuscript and responses to each individual question shortly this week.

Look forward to our further discussion.

Yours,

Authors

---

> ### Author Response · Authors · 2024-11-21
> **General responses (part 2)**
>
> Dear reviewers,
>
> Here, we continue to provide the responses to another common question raised by reviewer `oZCz`, `SYtw`, and `NVGL` regarding computational cost.
>
> ### **What is the computational cost of HESSO-(CRIC)?**
>
> We analyze the time and space complexity of HESSO-(CRIC) against standard optimizers. In conclusion, HESSO-(CRIC) indeed requires additional time and space complexities while the additions are negligible. In our numerous realistic applications besides the presented academic benchmarks, HESSO-(CRIC) are quite efficient, typically as efficient as standard training via vanilla optimizers.
>
> > **Table. Notations.**
> > |Symbol|Definition|Remark|
> > |--|--|--|
> > |$N$|# of trainable variables with gradient||
> > |$G$|The set of parameter groups.|The common setup could be pruning/erasing zero-invariant groups.|
> > |$\|G\|$|The size of $G$.|**Typically negligible compared to $N$, see the below table.**|
> > |$T$|# of training steps.||
> > |$T_{ht}$|# of hybrid training steps.|Set as $T_{ht}=T/10$ in our generic recipe. |
> > |$P$|# of pruning periods.|Set as $P=10$ in our generic recipe. |
> > |$S$|# of sampling steps in CRIC.|Set as $S=10$ in our generic recipe.|
> > |$C$|# of cycles in CRIC.|Empirically terminates within about 10 cycles.|
>
> > **Table. Magnitude Comparison Between $N$ and $|G|$.**
> > |Model|$N$|$\|G\|$|Ratio $\|G\|/N$|
> > |--|--|--|--|
> > |CARNx2|$9.6*10^5$|$1.7*10^3$|$1.8*10^{-3}$|
> > |ResNet50|$2.6*10^7$|$1.2*10^4$|$4.6*10^{-4}$|
> > |Yolov5-Large|$4.7*10^7$|$2.6*10^4$|$5.5*10^{-4}$|
> > |Bert-Base|$1.1*10^{8}$|$3.8*10^4$|$3.5*10^{-4}$|
> > |Phi2-2.7B|$2.7*10^9$|$4.1*10^5$|$1.5*10^{-4}$|
>
> > **Table. Space and Time Complexity Comparison.**
> > |**Optimizer**|**Variant**|**Space Complexity (Peak)**|**Time Complexity**|**Space Complexity Projected onto Phi2**|**Time Complexity Projected onto Phi2**|
> > |--|--|--|--|--|--|
> > |SGD|Standard|$O(2N)$|$O(NT)$|$O(2N)$|$O(NT)$|
> > |**HESSO**|**SGD**|$O(2N+\|G\|)$|$O(NT+\|G\|T_{ht}+\|G\|P)$|$O(2.00015N)$|$O(1.000015NT+1.5*10^{-3}N)$|
> > |**HESSO-CRIC**|**SGD**|$O(2N+\|G\|S)$|$O(NT+\|G\|T_{ht}+\|G\|SC)$|$O(2.0015N)$|$O(1.000015NT+1.5*10^{-1}N)$|
> > |Adam/AdamW|Standard|$O(3N)$|$O(2NT)$|$O(3N)$|$O(2NT)$|
> > |**HESSO**|**Adam/AdamW**|$O(3N+\|G\|)$|$O(2NT+\|G\|T_{ht}+\|G\|P)$|$O(3.00015N)$|$O(2.000015NT+1.5*10^{-3}N)$|
> > |**HESSO-CRIC**|**Adam/AdamW**|$O(3N+\|G\|S)$|$O(2NT+\|G\|T_{ht}+\|G\|SC)$|$O(3.0015N)$|$O(2.000015NT+1.5*10^{-1}N)$|
> >
> > Remark: Compared to SGD, Adam and AdamW require additional storage for the second-order momentum of the gradient and adjust the search direction based on this momentum.
>
> Yours,
>
> Authors

---

### Note · Authors · 2025-01-02

I have read and agree with the venue's withdrawal policy on behalf of myself and my co-authors.